# Assessing the Stability of Surface Lights for use in Retrievals of Nocturnal Atmospheric Parameters

Jeremy E. Solbrig[1], Steven D. Miller[1], Jianglong Zhang[2], Lewis Grasso[1], Anton Kliewer[1]

[1]Cooperative Institute for Research in the Atmosphere, Fort Collins, CO, 80523, USA

[2]Department of Atmospheric Sciences, Grand Forks, ND, 58202, USA

*Correspondence to*: Jeremy E. Solbrig (jeremy.solbrig@colostate.edu)

**Abstract.** Detection and characterization of aerosols is inherently limited at night because the important information provided by visible spectrum observations is not available and infrared bands have limited sensitivity to aerosols. The VIIRS Day/Night Band (DNB) onboard the Suomi-NPP satellite is a first-of-its-kind calibrated sensor capable of collecting visible/near-infrared

observations during both day and night. Multiple studies have suggested that anthropogenic light emissions such as those from cities and gas flares may be useable as light sources for retrieval of atmospheric properties including cloud and aerosol optical depth. However, their use in this capacity requires proper characterization of their intrinsic variation, which represents a source of retrieval uncertainty. In this study we use 18 months of cloud-cleared VIIRS data collected over five selected geographic domains to assess the stability of anthropogenic light emissions and their response to varied satellite and lunar geometries.

Timeseries are developed for each location in each domain for DNB radiance, four infrared channels, and satellite and lunar geometric variables, and spatially-resolved correlation coefficients are computed between DNB radiance and each of the other variables. This analysis finds that while many emissive light sources are too unstable to be used reliably for atmospheric retrievals, some sources exhibit a sufficient stability (relative standard deviation < 20%). Additionally, we find that while the radiance variability of surrounding surfaces (i.e. unpopulated land and ocean) is largely dependent on lunar geometry, the

anthropogenic light sources are more strongly correlated to satellite viewing geometry. Understanding the spatially-resolved relationships between DNB radiance and other parameters is a necessary first step towards characterizing anthropogenic light emissions and establishes a framework for a model to describe variability in a more general sense.

## 1 Introduction

Atmospheric aerosols have wide-ranging impacts across the globe. These impacts occur on both local and global scales, over

both the short- and long-term, and are both environmental and biological. The impact of aerosols on global climate is well known (e.g. Hansen et al., 1992). On shorter time-scales, the presence of aerosols impacts cloud microphysical processes (e.g. Kaufman et al., 2002) and their subsidence onto snow surfaces can lead to increased rate of snow melt, decreased snow cover, and increased ocean turbidity (e.g. Painter et al., 2007). Biologically, aerosols are known to have adverse health impacts resulting in higher rates of asthma and other pulmonary diseases and have been linked to shorter lifespan (e.g. Silva et al.,

2013).  Commercial and military operations are frequently impacted by reduced visibility due to dense aerosol plumes and require information on when and where visibility is expected to be reduced (e.g. Zhang et al., 2008).

Despite the wide-ranging impacts of aerosols our ability to detect plumes is largely limited to the daytime when we can employ
techniques that rely on visible light such as the Dark Target (Kaufman et al., 1997; Levy et al., 2007; Remer et al., 2005), the Deep Blue (e.g. Hsu et al., 2013), and the Multiangle Implementation for Atmospheric Correction (MAIAC; Lyapustin et al., 2011a, 2011b, 2012) methods.  While some active sensors like the Cloud/Aerosol Lidar with Orthogonal Polarization (CALIOP; Winker et al., 2003) are able to make retrievals of atmospheric aerosol optical depth (AOD) at night, they have limited spatial coverage because they are limited to a vertical "curtain" of data.  A small number of algorithms such as the
Dynamic Enhancement Background Reduction Algorithm (DEBRA) dust algorithm (Miller et al., 2017) are able to perform limited aerosol detection at night using infrared techniques, however, these algorithms are limited to plumes containing larger particles and are qualitative in nature.

The launch of the Visible Infrared Imager Radiometer Suite (VIIRS) onboard the Suomi National Polar-orbiting Partnership
(Suomi-NPP) and Joint Polar Satellite System-1 (JPSS-1) satellites has extended observations in the visible spectrum from daytime into night.  VIIRS's Day/Night Band (DNB) (Lee et al., 2010; Miller et al., 2013; Solbrig et al., 2013) is a broad-band visible to near-infrared radiometer with multiple gain settings allowing it to produce consistent imagery from daytime, through the terminator, into nighttime.  Its broad dynamic range allows the DNB to image the brightest cities without saturating while also observing clouds on moonless nights using only the light emitted by the atmosphere itself (Miller et al., 2015).  Unlike its
predecessor, the Operational Linescan System (OLS) on the Defense Meteorological Satellite Program (DMSP) platforms, the DNB is calibrated and can be used for quantitative applications.  DNB imagery has proven useful for many applications including cloud property retrievals (Walther et al., 2013), detection of low-altitude circulation centres in tropical cyclones (Hawkins et al., 2017), forest fire detection and estimation of fire radiative power (Elvidge et al., 2013; Polivka et al., 2016), lightning detection (Bankert et al., 2011), identification of fishing boats and fishery monitoring (Elvidge et al., 2015 and 2018),
mapping power outages in the wake of natural disasters (Miller et al., 2015 and 2018; Molthan and Jedlovec, 2013), and monitoring socioeconomic activity (Ma et al., 2014).

The DNB's ability to measure radiance emitted by city lights may also prove useful for development of algorithms that use city lights as a known light source.  Several studies have focused on using city lights as a light source for retrieval of nighttime
aerosol optical depth.  Zhang et al. (2008) discuss the possibility of using artificial light sources to detect aerosol using both calibrated and uncalibrated observations of visible radiances and show application of this method to uncalibrated data from the OLS.  Building upon this method, Johnson et al. (2013) use calibrated data from VIIRS and find that the contrast between artificial light sources and nearby dark background locations can be used to retrieve AOD.  A variance-based method for retrieving AOD is presented by McHardy et al. (2015) which, unlike the method presented by Johnson et al. (2013), does not

require use of a nearby reference background location. Instead, the variance-based method computes the spatial standard deviation of observed brightness for individual cites on cloud/aerosol free nights, then compares this value against the standard deviation of brightness for the same cities on nights where aerosol is present. This method allows Zhang et al., (2019) to develop an automated method for deriving aerosol optical depth over cities globally.

If retrievals based on the use of city lights can be expanded globally they may provide a significant improvement in spatial coverage over the currently available algorithms. Although estimates of urban extent vary widely due to both definition and methodology, a reasonable estimate for total urban extent would be approximately 0.5% of Earth's total land area (approximately 700,000 km$^2$). If even 1% of anthropogenic light sources prove useful for use in aerosol retrievals they would

provide the largest available set of nighttime retrievals to date with. If 1% of all anthropogenic light sources are useful in retrieval algorithms those retrievals would cover approximately 0.3% of the globe. Comparatively, CALPSO, whose AOD retrieval currently provides the best nighttime spatial coverage, observes only 0.003% of the Earth daily.

If retrievals based on the use of city lights can be expanded globally, they may provide a significant improvement in spatial

coverage over currently available retrievals. Estimates of urban extent vary widely (Schneider et al., 2009; Doxsey-Whitfield, 2015). Given the estimate range, it might be reasonable to estimate that total urban extent is approximately 0.5% of Earth's land area (about 700,000 km2). If all of the urban area were useable in optical depth retrievals, a single VIIRS instrument, whose spatial resolution is 750 m, would be able to make about 1.2 million observations per night. If only 1% of the urban-lit area is usable the number would drop to about 12,000 observations per night. How much of the global urban area is useful for

optical depth retrievals will be algorithm and application dependent as well as dependent on how well the brightness of each light source can be constrained.

Each of these studies has pointed out the need for robust characterization of surface light sources to improve and quantify

uncertainty in their AOD retrieval methods. The studies indicate that the use of city lights in retrieval algorithms requires knowledge of their stability and how they vary under clear-sky conditions. In this study we examine DNB radiance observations to assess the stability of light sources and whether their variability can be attributed to known variables. We hypothesise that some sources of visible light at night will be sufficiently stable to be used in atmospheric retrievals while other sources will prove too unstable for use.

This analysis provides a first step towards characterization of anthropogenic light sources for use in retrievals and is to be agnostic of the retrieval algorithm. Some algorithms may be capable of performing retrievals using attenuation of light emitted by point sources. Others may rely on the amount of "blooming" observed around a light source to retrieve optical depth. It may also be possible to retrieve optical depth by observing changes in the brightness or spatial structure (e.g. spatial variance)

of groups of well-characterized light sources. Regardless of the algorithm employed, understanding the variability in anthropogenic light emissions and its causes is important to the problem of retrieving optical depth at night.

To assess the stability of nocturnal light sources we have gathered 18 months of VIIRS data over five different domains (the data and domains are described in Sect. 2). From these data we construct cloud-cleared timeseries at each location in the study domains for several variables including DNB radiance, brightness temperature data from four coincident infrared channels, and variables describing satellite and lunar geometry (described in Sect. 3). A basic analysis of the stability of nocturnal light sources is performed in Sect. 4. To assess the dependence of light source brightness on several variables we compute spatially resolved correlation coefficients between DNB radiance and the several other variables (Sect. 5). If done globally, constructing a model to describe how DNB radiance varies for each independent pixel based on other known variables would result in nearly one billion independent models. In Sect. 6 we summarize the results of this study, discuss conclusions, and present avenues for future research.

## 2 Satellite Data and Study Domains

### 2.1 VIIRS Data

The first VIIRS instrument, carried on board the Suomi-NPP satellite (Hillger et al., 2013; Lee et al., 2010), was launched on 28 October 2011. A second VIIRS now flies on the JPSS-1 satellite, launched on 18 November 2017. Both satellites fly in sun-synchronous orbit with a 1330 local time ascending node and a 0130 local time descending node. VIIRS provides quantitative imaging data in 22 traditional radiometric bands ranging from visible (0.412 μm) to longwave infrared (12.01 μm). Five of these channels, termed "Image" bands, have spatial resolution of 375 m at nadir and the other 17 channels, termed "Moderate" bands, have spatial resolution of 750 m at nadir. A novel element of VIIRS is a broadband low-light visible channel called the Day/Night Band (DNB) (Miller et al., 2013; Solbrig et al., 2013). The DNB is a broad-band visible to near-infrared (500-900 nm spectral bandpass) channel that is capable of measuring upwelling radiance during both day and night. The DNB provides near constant spatial resolution of 742 m across its 3000 km wide swath, preserving detail even near the scan edge.

For the current study, Soumi-NPP VIIRS data were gathered for all nights during 18-month period spanning January 2015 to June 2016 from the Comprehensive Large Array-Data Stewardship System (CLASS) maintained by the National Oceanographic and Atmospheric Administration (NOAA). The data gathered include DNB radiances, brightness temperature from four coincident infrared channels (channels 13, 14, 15, and 16; central wavelengths 4.05, 8.55, 10.76, and 12.01 μm respectively; see Table 1). Geolocation data were gathered for both the DNB resolution and the Moderate resolution infrared bands including latitudes, longitudes, satellite azimuth and zenith angle, and lunar azimuth and zenith angle. To assist with

the cloud-clearing quality control task, the VIIRS Cloud Mask (VCM; Kopp et al., 2014) Level-2 VIIRS product was gathered coincident with the DNB imagery. The VCM is discussed in more detail in Sect. 3.2.

## 2.2 Study Domains

This study concentrates on five different domains of interest, providing a diverse sampling of light sources and scene variability. Four of these domains are centred on cities and their surroundings including: the San Francisco Bay Area, CA, USA; Las Vegas, NV, USA; St. George, UT, USA; and Doha, Qatar. The fifth domain focuses on isolated sources of artificial light associated with oil wells located approximately 50 km west of Basra, Iraq. Associated with these oil wells are gas flares, used to burn off natural gas during oil extraction, which are hot and visibly bright when they appear as point sources (with blooming) in night time DNB imagery. The five domains are defined in Table 2 by their northwest and southeast corner coordinates in latitude and longitude. Each domain is 256 x 256 pixels and is remapped to stereographic projection at ~750 m spatial resolution (the nominal resolution of VIIRS data). The remapping is required for compilation of statistics for this analysis.

The rationale for selecting each of the domains is also described in Table 2. Each domain provides different types of anthropogenic light emissions including densely populated areas (e.g. San Francisco and Doha), small towns (e.g. St. George), areas of known industrial activity (e.g. Gas Flares domain), and areas of intense commercial activity (e.g. the Las Vegas "Strip"). Each domain provides a unique array of considerations and associated challenges, including frequent cloud contamination (e.g. San Francisco and Gas Flares domains), ephemeral lights from fishing boats and new construction (e.g. Doha and San Francisco), and light emissions associated with hot sources (e.g. Gas Flares and Doha).

## 3 Data Preparation

### 3.1 Construction of Databases

To study the stability and variability of DNB observed radiances, we constructed time series of DNB radiance at fixed locations. For each study domain, we compiled a database containing DNB radiance, four VIIRS infrared channels, the VCM, and several geometric variables (satellite zenith angle, satellite azimuth angle, lunar zenith angle, and lunar azimuth angle). Each dataset was configured as an $M$ x $N$ x $T$ temporal stack of two-dimensional spatial arrays where $M$ and $N$ are the latitudinal and longitudinal dimensions of the domain and $T$ is the number of satellite overpasses for the domain accumulated over the 18-month period.

The VIIRS data for a given study domain were co-registered using the nearest-neighbour interpolation method. While nearest-neighbour interpolation ignores the effects of the point spread function for each detector, it is used to avoid blending observations. The remapping is a source of error for this study but is necessary to create timeseries at each pixel. An analysis

conducted on other interpolation techniques (e.g. bilinear, cubic spline), not presented here, indicates that they mask, but do not remove, some of the error that is attributable to interpolation. The interpolation process results in a database containing timeseries for each VIIRS dataset at each pixel in our study domains.

## 3.2 Applying the VIIRS Cloud Mask

When developing robust baseline statistics on terrestrial artificial light source variability, we must consider only pixels observed under cloud-free and low-aerosol conditions. To this point, Figure 1 shows a relatively cloud-free image of the San Francisco Bay Area on 23 February 2015 (left) and a cloud-contaminated image from the same area on 27 January 2015, both collected at approximately 0200 local time. Both dates correspond to nights when moonlight was not contributing to the DNB imagery. The cloud-free image shows significant structure in the terrestrial light field with population centres, roads, and

bridges easily discerned. The cloud-contaminated image, on the other hand, gives only a rough impression of the shape of the emission sources. Note also the character of the surface lighting—sharply defined with significant spatial heterogeneity in the clear case, while dampened and blurred in the cloudy case. These impacts can be thought of as an extreme example of how atmospheric aerosol impacts the brightness and structure of artificial lights on a significantly reduced scale—forming the premise for the use of surface lights for atmospheric property retrievals. Given the high sensitivity of DNB radiances to the

presence of cloud, we must be careful to remove as much cloud contamination as possible so as not to bias the statistics.

As an initial screen for clouds, we use the operational VIIRS Cloud Mask (VCM) (Algorithm Theoretical Basis Document; https://www.star.nesdis.noaa.gov/jpss/documents/ATBD/D0001-M01-S01-011_JPSS_ATBD_VIIRS-Cloud-Mask_E.pdf). Using multiple radiometric tests, the VCM determines cloud probability on a per-pixel basis (Kopp et al., 2014). The cloud

mask is reported in terms of confidence levels (confident clear, probably clear, probably cloudy, or confident cloudy) and includes a flag indicating the quality of the mask ("High," "Medium," "Low," or "Poor") at each pixel. Also provided, but not used in this study, are the detailed results for each radiometric test used to construct the cloud mask.

Given the large amount of DNB radiance data available and the extreme importance of avoiding cloud contamination, we took

an aggressive approach to cloud screening with the VCM. All pixels that were not marked as both "Confident Clear" and "High" quality in the VCM were flagged in our dataset and excluded when calculating statistics and correlations. Doing so may remove some cloud-free data and reduce our sample size, but that is preferable to allowing cloud contamination in the dataset.

Although we have adopted an aggressive posture to the implementation of VCM filtering, this does not guarantee that all clouds will be eliminated from the remaining DNB data. Figure 2 illustrates this challenge, showing two example DNB radiance images over the San Francisco Bay Area where the VCM has been applied (colour-coded in blue). Qualitatively, the VCM appears to perform well in the left-hand image, but performs poorly in the right-hand image, as indicated by the tell-tale

blurring of the terrestrial light sources by undetected clouds. Similar poor performance was observed frequently throughout the 18 months of data and occurred to varying degrees in each of the study domains. Evaluation of nighttime VCM performance, based on the incorporation of DNB information and artificial light (in addition to moonlight) is a topic ripe for follow-on research.

**3.3 Manual Cloud Screening**

As an additional step to mitigating the issues of clouds left undetected by the VCM, we conducted a manual examination of the VCM-filtered DNB radiance imagery for each overpass. The variance in visible radiance has been used for nearly two decades as a method of screening residual cloud. For example, Martins et al. (2002) expand upon the MODIS cloud mask for use in aerosol applications by masking regions of high spatial variance in visible brightness as cloud. Conversely, using expert

analysis, we identified any overpasses containing significant reductions in spatial variance of the terrestrial light sources (e.g. Figure 2, right), indicating the presence of cloud or aerosol that was not masked by the VCM. The data for these cloud-contaminated overpasses were removed from our ensuing analysis. Note that this process will also remove scenes that are heavily polluted by aerosols as they impart a similar blurring effect. The variance in visible radiance has been used for decades as a method of screening residual cloud (e.g. Martins et al., 2002), albeit . This results in a reduced, but higher-quality dataset.

The number of scenes remaining after the manual cloud-screen is shown in the last column of Table 2 for each of the five domains. Most of the domains have between 200 and 350 scenes, however, the Iraqi Gas Flares domain is reduced to only 95 scenes which limited some analysis of this study.

Note that, while this method helps screen substantially cloudy scenes, it is an all-or-nothing approach. If a significant amount

of blurring is observed in an image, that entire image was removed. However, if by subjective measure the amount of blurring present was deemed to not be "significant," the image was retained. As a consequence, some residual cloud may be present in the remaining data. While this manual screening method was deemed necessary for the current study, it is time consuming and not completely accurate. As such, this method is not ideal for processing larger datasets, and our methods will need to be improved in the future for bulk (e.g., global) processing.

**4 Day/Night Band Radiance Stability**

Using the databases constructed for each study domain, as per Sect. 3, we assessed the time-averaged characteristics of quality-controlled DNB radiances at the pixel level. We began by considering basic statistics of DNB radiance in each of our five domains and discuss their implications to aerosol characterization. Namely, we calculated the minimum, maximum, mean, and relative standard deviation (RSD) of the DNB radiance at each pixel in each domain (e.g. Figure 3). The results are

30 presented as two-dimensional images for each form of statistic to preserve the spatial character of the source information.

In the "minimum" radiance images anthropogenic light sources that are not transient appear bright while transient light sources appear dim or dark. Areas far removed from anthropogenic light sources appear dark due to a lack of moonlight on new moon nights. The immediate surroundings of anthropogenic light sources exhibit some visible signal due to scattering of anthropogenic light from the surface.

Anthropogenic light sources in the "maximum" radiance images are at their brightest. Both transient and non-transient light sources appear bright. Areas far removed from anthropogenic light sources appear dimmer than most anthropogenic light sources, but still are relatively bright due to surface-reflected moonlight. These areas appear approximately the same as daytime visible imagery with clouds removed. The areas immediately surrounding anthropogenic light sources appear the same as the surrounding land since the surface-scattered anthropogenic light is orders of magnitude dimmer than the reflected moonlight.

The "average" radiance imagery gives an idea of the average brightness at each pixel over the time-series. The brightness of each pixel is between the brightness of the corresponding pixels from the "minimum" and "maximum" radiance images.

While interpretation of the minimum, maximum, and average radiance images is straightforward, RSD, which is common in statistics but non-standard in atmospheric science, benefits from definition here. For our application, RSD provides a measure of stability over time at a given location. It is defined as the standard deviation of the DNB radiance divided by the average DNB radiance ($RSD = \frac{\sigma}{\mu}$) evaluated at each pixel. As such, an RSD = 1.0 indicates that the standard deviation is equal to the mean, an RSD < 1 indicates that a pixel is relatively stable over time since its standard deviation is lower than its mean value, and RSD > 1 indicates that a pixel is more variable over time because the standard deviation of its radiance is greater than its mean value. Given this definition, pixels away from anthropogenic light sources have RSD >> 1 due to changes in brightness during the lunar cycle. Stable visible light sources have RSD << 1 and transient or variable visible light sources have higher RSD values. Areas immediately surrounding visible light sources have relatively low RSD compared to pixels farther from the light sources due to surface scatter of the nearby emitted light.

While simplistic in construct, these statistics combine to form a suite of quantitative information useful for understanding the character of light emissions for each of our analysis domains. Considered together, they provide information vital to determining: 1) which locations are potentially most suitable for use as stable light sources, 2) gauging the amount of residual cloud contamination present in the cloud screened dataset, and 3) determining which locations contain transient/ephemeral light sources (e.g. fishing boats and flashing/strobing lights) which would produce significant and spurious aerosol retrieval uncertainties.

## 4.1 Radiance Statistics: Qatar

Qatar, a small peninsular country bordering Saudi Arabia and situated on the southern coast of the Persian/Arabian Gulf, was chosen for its relatively high number of cloud-free nights and diversity of artificial light sources. Light sources in Qatar include the dense urban area of Doha (population ~633,000) on the eastern coast, many smaller towns scattered across the peninsula, long stretches of isolated, well-lit highways, frequent fleets of fishing boats near its ports, and significant gas flaring from nearby oil wells. Figure 3 shows the (a) minimum, (b) maximum, (c) average, and (d) RSD of DNB radiances at each location in the Qatar domain.

The standard measures of minimum, maximum, and average radiance give immediate insight on the attributes of artificial light sources in the Qatar domain. Locations far-removed from artificial light sources (hereafter called "unpopulated" locations) exhibit very low minimum DNB radiance, but are three to four orders of magnitude brighter at their maximum (i.e. they have a large dynamic range of radiance). Without the presence of anthropogenic light sources, the variability of unpopulated locations is tied primarily to lunar illumination and thus bears a strong dependence on lunar geometry and phase. In contrast, regions of high radiance (i.e. artificial light sources; hereafter called "populated locations" in the DNB imagery) produce high values of minimum, maximum, and average radiance. The brightness of most populated locations remains stable (i.e. having similar brightness at minimum and maximum) in comparison with unpopulated locations. This stability indicates that the brightness of populated locations has relatively low dependence on lunar phase and lunar geometry, as would be expected for a source whose brightness is significantly larger than the lunar variability. These relationships are discussed in more depth in Sect. 5.1.

In the minimum radiance example (Figure 3a), while some blooming can be seen around the brighter light sources due to atmospheric scattering (light dome) and second-order scatter from the light dome and the underlying surface, the structure of the city lights themselves remains relatively stable when compared to both the average and maximum radiance composite imagery. This indicates that there are few instances where cloud or aerosol obscures the domain, and our cloud screening protocol has been mostly successful. Regardless, cloud and aerosol contamination remain a likely source of error in the clear-sky radiance statistics due to imperfect screening as discussed.

Transient and semi-transient light sources appear in the maximum and average radiance images, but do not appear in the minimum radiance image. Off the eastern coast of Qatar are two regions of sparse lights that appear in both the average and maximum radiance composite imagery, but these regions are absent in the minimum radiance composite image. These sources are most likely related to boat activity. Likewise, the "Orbital Highway" (blue arrow in Figure 3b) appears in both the maximum and average images but is absent in the minimum image. The absence is most likely because the Orbital Highway was under construction between 2015 and 2017 and did not exist in the beginning of our time series. Just south of Qatar's

border with Saudi Arabia a bright light source is present in the maximum image but is mostly absent in the average image and completely absent in the minimum image. Examination of infrared imagery (not shown) indicates that this is likely an intermittently operating oil well gas flare, emitting a large, bright flame. Hence, the stability data can capture and omit from consideration these transient light sources as candidates for atmospheric parameter retrievals.

Figure 3d shows the RSD of DNB radiance at each location, which gives a measure of the stability of the light sources as described in Sect. 4. Here, we show RSD in the range from 0.0 to 1.0 where a value of 0.0 indicates that a location's radiance is always exactly the same (i.e. perfectly stable in all conditions) and a value of 1.0 indicates that $\sigma = \mu$. Many locations have RSD > 1.0 and are shown in black. Such highly variable locations are deemed of lesser value for atmospheric parameter retrievals.

In Figure 3d the "transient" light sources such as fishing boats, the recently constructed Orbital Highway, and gas flares are expectedly absent. More stable light sources can be recognized as well-lit highways, towns, and dense urban centres. Some of the remaining, less transient light sources are remarkably stable, with RSD < 0.2 (e.g. the centre of Doha) while others are less stable with RSD > 0.5 (e.g. suburbs, towns, and highways). Thus, the RSD metric provides a first-order quantitative indication of which light sources are most likely to be useful in atmospheric parameter retrievals and may be further analysed to characterize retrieval uncertainties tied to their natural variability.

## 4.2 Radiance Statistics: Las Vegas, NV

Like Qatar, the desert climate of Las Vegas, NV provides a high number of cloud free nights in the time series. It also offers a diverse set of terrestrial light sources, including the larger metropolitan area (city population ~620,000; metro population ~2.1 million), the bright and structurally complex "Las Vegas Strip," the nearby moderate-sized town of Pahrump, NV (population ~36,000), and many smaller towns in the Las Vegas suburbs. Following the same model of analysis as Qatar, Figure 4 shows the (a) minimum, (b) maximum, (c) average, and (d) RSD of DNB observed radiance composite imagery for the Las Vegas domain.

Generally, DNB radiances in this domain exhibit similar behaviour to those in the Qatar domain. Once again, unpopulated locations have low minimum radiance (Figure 4a) but are three to four orders of magnitude brighter at maximum (Figure 4b), tied to lunar geometry. Populated locations appear to be relatively stable between the minimum, maximum, and average (Figure 4c) radiance images. Cloud clearing appears to have performed well, as indicated by the lack of strong blooming (caused by cloud-scattered light) in the maximum radiance image.

As with Qatar, the Las Vegas domain features light sources having variability idiosyncrasies. As an extreme example, the Las Vegas "Strip," known for its brightly lit casinos and hotels, is located in the centre of the city. The Strip is extremely bright

in the minimum, maximum, and average images indicating that it might be a good candidate for use in atmospheric retrievals. However, the RSD image (Figure 4d) shows that the Strip is also highly variable, with RSD > 1.0 (the most variable location has RSD = 3.2). This is likely caused by the prevalence of flashing, strobing, or otherwise changing light sources on the Strip. This indicates that, while much of the Las Vegas area is likely to be stable and potentially useful for retrievals, the Strip itself, while very bright, is inherently too variable to be used in that capacity.

## 4.3 Radiance Statistics: St. George, UT

Like the previous two domains, the St. George, UT domain, located on the northern extent of the Mojave Desert, was selected in part due to its favourable cloud climatology. In contrast to the domains discussed in previous sections, however, the city of St. George is relatively small (city population ~85,000; metro population ~165,000) and has little vertical structural development (e.g., skyscrapers).

Statistics for the St. George domain are shown in Figure 5. The lack of blurring in the minimum and maximum images once again indicates that cloud screening was effective. Despite the significant difference in population compared to the previous domains, the regions of this domain that contain artificial lights appear to vary little between the minimum, maximum, and average images. This is also seen in the RSD image which shows relatively low RSD, as low as 0.1 in some locations within St. George. Otherwise, the St. George domain is unremarkable compared to the previously considered domains.

## 4.4 Radiance Statistics: San Francisco, CA

San Francisco provides a much more complex nocturnal artificial lights environment than St. George. While, like Qatar and Las Vegas in terms of its varied light source types, it is also significantly and regularly impacted by cloud contamination. Marine stratocumulus, frequent in the area, is difficult for the infrared-based VCM to detect at night, especially over land owing to poor thermal contrast between the cloud and the complexities of the underlying surface. Thus, cloud contamination presents a greater challenge for this domain.

Figure 6 shows statistics for the San Francisco Bay Area. While the behaviour is, in general, similar to the other domains presented so far, there is one major difference—the adverse effects of cloud contamination (seen in the minimum, maximum, and average statistics). For example, many artificial lights that are visible in the maximum radiance image are dimmer or entirely absent in the minimum radiance image (due to optically thick, yet undetected, cloud obscuration). The maximum brightness image appears blurry, indicating that scattering clouds are sometimes present. These artefacts occur despite our best efforts at cloud screening using both the VCM and the "all-or nothing" manual methods (see Sect. 3.2 for a discussion of the issues with the VCM and 3.3 for a discussion of the manual screening method). The cloud impacts are especially evident in the maximum radiance image over the northern tip of the San Francisco Peninsula and in the relatively unpopulated region

between Half Moon Bay and San Francisco (indicated by blue arrow). The cloud contamination problem points to the need for longer-term sampling to build more robust clear sky statistics in these climatologically cloudy zones.

Easily seen in Fig. 6 are the four major bridges crossing the bay (from north to south: Richmond – San Rafel Bridge, San Francisco – Oakland Bay Bridge, San Mateo – Hayward Bridge, and Dumbarton Bridge). Each of these bridges, being well-lit highways, are both bright and moderately stable (RSD < 0.5). In contrast, the Golden Gate Bridge, which crosses the western mouth of the San Francisco Bay, appears bright but is more unstable source (RSD > 1.0). Dense population centres (e.g. downtown San Francisco, San Jose, Oakland, etc.) are far more stable (RSD < 0.2) and thus are better candidates for use in atmospheric parameter retrieval algorithms. As with previous examples, suburban areas are less stable and thus less useful for use in retrievals.

## 4.5 Radiance Statistics: Iraqi Gas Flares

The final case domain examines an oil field in south-eastern Iraq. A line of bright light point sources (marked in blue in Figure 7e) centred approximately 50 km west of Basra are the result of the controlled burn-off of natural gas from oil wells. These flares appear consistently bright in DNB radiance. In addition to being bright at DNB wavelengths, they also register as hot in the VIIRS infrared channels (not shown) with typical physical temperatures of approximately 1800 K (Elvidge et al., 2016). While this domain does contain some light emissions from cities (e.g. Basra) we are more interested in the stability of the gas flares as point sources for atmospheric characterization.

Cloud screening for this domain was particularly troublesome. Hot locations like gas flares inadvertently trigger one of the VCM cloud tests, causing false alarms for cloud. Consequently, the gas flares are marked as "cloud" almost 100% of the time in the VCM. To mitigate this, we introduced a restoral test based on the difference of the 4 μm and 11 μm infrared brightness temperature ($BTD_{4-11}$). While some transient features can exhibit a positive $BTD_{4-11}$, only a hot stationary object should consistently produce a strong positive value in the same location. We calculate the time-averaged $BTD_{4-11}$ for each location in the domain. Any location whose time-averaged $BTD_{4-11}$ is positive is classified as a gas flare. Additionally, to capture the full extent of the masking error from the VCM, we unmasked any locations in the 5 x 5 box surrounding any detected hot locations.

While the restoral process unmasks the gas flares in all overpasses, regardless of cloud cover, we attempt to remove any scenes containing cloud during our manual examination of the imagery (Sect. 3.3). We were extremely strict during our manual screen for this domain, removing over 70% of all scenes due to expert assessment that cloud and/or significant aerosol contamination was present. Comparison of the minimum and maximum DNB radiance statistics for each location in this domain (Figure 7 a and b) shows that for both the gas flares and the city lights of Basra the DNB radiance varies significantly

in time. Additionally, there is significant blooming of the light from the gas flares in the average DNB radiance image (Figure 7c). This indicates that, despite our best efforts, cloud screening may have been insufficient to fully quality control this domain.

Despite the cloud mask limitations and the significantly reduced sample size, this domain provides useful information about the stability of gas flares as light sources. In the RSD image (Figure 7d) we can see that the light emitted by Basra is relatively stable (RSD values of 0.2-0.3). The oil wells, on the other hand, have RSD values approaching 2.0 (colour bar only extends to 1.0) indicating that they are extremely variable and, unless correlations with predictive variables can be found, would not be useful in retrievals. As noted previously, however, while the use of Nearest Neighbour interpolation leads to a more accurate portrayal of the DNB observations, it may introduce variability in some situations, especially in the case of isolated sub-pixel light sources such as gas flares. Our choice of interpolation scheme is a likely cause of some of the variability observed in this domain but, as discussed in Appendix A, use of a higher order interpolation scheme would smooth the resulting imagery and remove the real impact of sensor optical properties from the results. There is still the potential that these light sources may be useful if consistent relationships can be established between their visible brightness and their physical temperature (correlation with the infrared bands).

## 5 Radiance Correlations

As seen in the previous section, even the most stable light sources exhibit variability with RSD near 0.1. While some of the variability may be explained by sources of error within our analysis (e.g. incomplete cloud and aerosol screening and interpolation to a standard projection), some of the variability may be explained by relationships between DNB radiance and other known parameters. If true, variability could be further reduced by introducing constraints on these parameters. In this section, we examine the relationships that exist between DNB radiance and various other observable parameters, including infrared brightness temperature, satellite viewing geometry, and lunar geometry. We note that since the satellite overpasses each domain at approximately the same time each night, the lunar phase is implicit to the lunar zenith angle. Also presented here are the relationships between DNB radiance and each of the *a priori* variables expressed as the correlation coefficient at each pixel. Here, as we use the Pearson Correlation Coefficient, we assume that our relationships are linear and, while this may not always be the case, this allows us to begin to explain some of the variability in DNB radiance. Further research will be required to develop more robust relationships.

It was determined from the above analyses that the Iraqi Oil Wells domain is unreliable, due to cloud screening resulting in a significantly reduced dataset. The sample size for this domain should be considered to be too small for significant conclusions to be drawn. In fact, the small sample size results in data artefacts in these correlations. The Iraqi Oil Wells domain is included here to facilitate discussion of correlation between infrared brightness temperature and DNB radiance and for completeness.

We begin by discussing the most readily explained relationship, between DNB radiance and lunar zenith angle, and work towards relationships of increasing complexity. The results below are shown for all domains including Iraq, however, the statistical strength for the Iraq domain is weaker due to smaller sample size for reasons discussed in Section 4.5.

## 5.1 Lunar Zenith Angle Dependency

The most straightforward relationship from which to draw conclusions is that of DNB radiance with lunar zenith angle (LZ) which is 0° when the moon is overhead and 180° when the moon is directly behind the Earth. Examining this relationship can give us confidence that our method for calculating correlations is working as intended. By the nature of Suomi-NPP's sun-synchronous orbit and its 0130 local overpass time, the LZ can also be seen as a proxy for lunar phase with lower values of lunar zenith angle corresponding to higher fraction of illumination (i.e. towards full moon). As will be discussed in Sect. 5.3,

LZ is not a perfect proxy for lunar phase, but for now, we will hypothesise that locations that do not contain an artificial light source should be brighter under low LZ than higher LZ, leading to negative correlations between DNB radiance and LZ. Additionally, since the emissions of larger cities are orders of magnitude brighter than moonlit surface reflectance, locations containing dense city lights should exhibit little to no correlation with LZ (i.e. correlation coefficient ≈ 0).

The spatially-resolved correlation coefficients for DNB radiance with LZ are shown in Figure 8 for all five study domains. In these figures, negative (red) values indicate that the DNB radiance and LZ are negatively correlated, meaning that DNB radiance decreases when the moon is lower in the sky (higher LZ) and at lower (e.g. quarter to Gibbous) phase. Zero (white) values indicate that the DNB radiance and LZ are uncorrelated. Positive (blue) values, which expectedly occur infrequently in this relationship, indicate that the DNB radiance and LZ are positively correlated such that the DNB radiance is higher when

the moon is below the horizon.

As expected, each of the five domains shows a strong negative (red) correlation between the DNB radiances and LZ in unpopulated areas (areas without strong anthropogenic light emission). This indicates that these locations are brighter when the moon is overhead than when the moon as at or below the horizon. Additionally, with the exception of the Iraqi Oil Wells

domain, areas with strong anthropogenic light emissions show little to no (white) correlation between DNB radiance and LZ. This lack of correlation is due to anthropogenic light emissions often being orders of magnitude brighter than reflected moonlight, even at lunar maximum. As noted previously, the Iraqi Oil Wells domain is composed of a small sample size and the statistics derived from its data are unreliable.

## 5.2 Satellite Zenith Angle Dependency

The relationship between DNB radiance and the satellite zenith angle (SZ) has more significant implications for the use of anthropogenic light emissions in retrieval algorithms. Figure 9 depicts the correlation coefficient for DNB radiance and SZ for the five study domains. In these figures, negative (red) correlation indicates that a location's DNB radiance is higher on

average when SZ is low (i.e. the satellite observes the location from overhead). Zero (white) correlation indicates that a location's radiance is the same on average regardless of whether it is viewed from above or from an oblique angle. Positive (blue) correlation indicates that a location's DNB radiance is higher when the satellite observes a location from an oblique angle (i.e. the satellite is either to the east or west of the location). Note that this formulation of SZ does not include east/west
viewing direction, only the absolute value of the viewing angle (directionality is discussed in Sect. 5.3).

Examining the unpopulated locations in each of the five domains shows varied relationships. For example, the land area surrounding Qatar (Figure 9a), San Francisco (Figure 9b), and the Iraqi Oil Wells (Figure 9e) shows a slight positive correlation between DNB radiance and SZ indicating that these locations are, on average, brighter when viewed from an oblique angle
than when viewed from directly above. In the Las Vegas domain (Figure 9c) there is a slight negative correlation indicating that the land surface in this domain is on average brighter when viewed from above than when viewed from an oblique angle. These differences may be indicative of differences in the scattering properties of the land surface in each of these domains. This is further evidenced by the two regions to the north of Las Vegas that show stronger negative correlation than the rest of the domain, indicating that they may have different scattering properties. The land surface in the St. George domain (Figure
9d) has both small positive and negative correlations with SZ due to the rugged and varied terrain surrounding St. George.

In all domains water shows a positive correlation to SZ. Near sources of anthropogenic light emission this positive correlation can be explained by reflection of the anthropogenic light off of the water surface towards the satellite. However, water surfaces more removed from populated areas still exhibit some positive correlation between DNB radiance and SZ. This effect is
discussed in Section 5.3.

More to the point for future retrieval algorithms, populated locations exhibit both positive and negative relationships. Positive relationships, which indicate that a location appears brighter when viewed from an oblique angle, are seen in areas with less vertical building development such as St. George and the outlying areas of larger cities. More densely populated areas such
as portions of Doha, the Financial District in San Francisco, and the Las Vegas Strip have negative correlations indicating that they appear brighter when viewed from above. These differences are likely due to a number of different factors including a higher prevalence of vertical emissions from buildings, street lights, and advertisements in dense areas and tall buildings obscuring light when viewed from oblique angles.

Other anthropogenic features that appear brighter when viewed from above include those that contain significant downward pointing light sources. These include the lighted highways that crisscross Qatar (linear red features in Figure 9a), bridges across the San Francisco Bay, large parking lots like those in strip malls (blue arrows in Figure 9c), and large industrial and commercial facilities (e.g. airports).

Figure 10 shows some interesting features from the San Francisco domain. Circle "a" in the left-hand zoom shows San Francisco's Financial District which has a strong negative correlation between DNB radiance and SZ indicating that it appears brighter when viewed from above. This is likely because the district contains many tall buildings that obscure emitted light when the district is viewed from oblique angles. Likewise, in the right-hand zoom, circle "e" indicates downtown San Jose

which exhibits similar characteristics. Circles "b" and "c" indicate areas that contain significant numbers of shopping centres with large parking lots that are lit by vertically oriented lamps. Circle "d" indicates San Jose International Airport, which is interesting because it shows little correlation between DNB radiance and SZ as well as high RSD (Figure 6d), likely due to the continually changing illumination at the active airport.

## 5.3 Directional Satellite Zenith Angle Dependency

The relationship between DNB radiance and SZ gave information on whether a location is brighter when viewed from above or from an oblique angle but is unable to tell us anything about the across-track directionality of a light source. To examine whether there is a dependence between DNB radiance and the direction from which a location is viewed when viewed from an oblique angle, we adjust the satellite zenith angle to create the "directional satellite zenith angle" (DSZ). While satellite azimuth angle might be used to fill this role, its results are difficult to interpret and result in more non-linear relationships. The

DSZ ranges from -70° to 70° and is defined such that when the satellite is to the east of the location it is viewing (i.e. the satellite is looking westward across its track) the DSZ is positive and when the satellite is to the west of the location it is viewing (i.e. the satellite is looking eastward across its track) the DSZ is negative. Given this definition for DSZ positive correlation between DNB radiance and DSZ indicates that a location is brighter when the satellite is positioned to the location's east (looking west) and negative correlation indicates that a location is brighter when the satellite is positioned to the location's

west (looking east).

Correlation between DNB radiance and DSZ is shown for each of the five domains in Figure 11. It is immediately obvious from these figures that many locations are brighter when viewed from one direction than when viewed from the other. While, for this study, we are most interested in the response of populated locations to viewing conditions, it is difficult to draw specific

conclusions from city locations due to their high spatial variability in this relationship. Directionality of emission and blocking of light due to buildings makes cities difficult to directly interpret but is important if city lights are to be used as light sources for retrievals.

Unpopulated land and ocean surfaces are much easier to interpret. For example, the land surface on the eastern (western) side

of a bright light source is brighter when the satellite is east (west) of the location it is viewing. This is most obvious in the unpopulated locations to the east and west of Las Vegas (Figure 11c). Locations to the east of Las Vegas are brightest when viewed from the east (blue) and locations to the west of Las Vegas are brightest when viewed from the west (red). This indicates that light emitted by the city is forward-scattered by the land surface. Another example of this can be seen in the unpopulated

locations surrounding the highway that runs northwards from Doha, Qatar to the northern tip of the peninsula (Figure 11a). Locations immediately east of the highway are brighter when viewed from the east (blue) while locations immediately west of the highway are brighter when viewed from the west (red) and the locations that contain the highway itself show no correlation with DSZ (white). Similar features can be seen around most population centres, with locations to the east (west) of the populated area brighter when viewed from the east (west) and the population centres themselves a mix of different relationships.

Another notable feature is a general negative (i.e. brightest when viewed from the west) correlation of both land and ocean features. With the exception of areas near populated locations (and the St George, UT domain, which is very rugged), unpopulated land and ocean tends to be brighter when viewed from the west. This can be explained by considering lunar geometry and the orbit of Suomi-NPP. Since Suomi-NPP views each location at approximately 0130 local time a larger portion of the moon will be illuminated when the moon is on the eastern horizon than when it is on the western horizon. This combined with preferential forward scattering from the surface means that more lunar illumination reaches the sensor when the sensor is to the east of a flat, unpopulated location. This is further illustrated by Figure 12 which shows correlation between DNB radiance and DSZ for only moonless nights (SZ > 95°). In this figure, without lunar illumination, the general tendency for unpopulated locations to be brighter when viewed from the west disappears. The correlation between DNB radiance and DSZ for locations far from population centres (e.g. the south-west corner of the San Francisco domain) is weak. The weakness of the correlation in these locations can be attributed to their radiance values which approach the noise floor for the sensor.

As noted previously, the unpopulated locations in the St. George, UT domain (Figure 11d) do not exhibit the same trend as the unpopulated locations from other domains. This is likely due to a combination of rugged terrain and, possibly, different scattering properties of the surface. Another area that breaks from the general trend for unpopulated locations is the northern tip of the Southern Coast Ranges located to the east of San Jose, CA in Figure 11b. As seen in Figure 12b, under moonless conditions this region is brighter when viewed from the east. This is likely due to scattering of light from the Bay Area cities off of the mountainous land surface. The opposite phenomenon can be seen to the west of the San Jose area.

**5.4 Infrared Brightness Temperature Dependency**

While most anthropogenic light sources likely exhibit little correlation between DNB radiance and physical temperature, some emission sources, such as flames emanating from oil wells and refineries, can be extremely hot. Elvidge et al. (2016) estimate that the physical temperature of oil well gas flares is typically about 1800 K. Consequentially, these light sources likely exhibit strong correlation between DNB radiance and infrared brightness temperature. Correlations were examined between DNB radiance and brightness temperature from four infrared channels (4.05, 8.55, 10.76, and 12.01 μm) and the details are shown in Table 1. Figure 13 and Figure 14 show the correlation coefficient for DNB radiance versus the 4.05 μm and 12.01 μm images, respectively. The correlations shown in Figure 13 and Figure 14 are remarkably similar despite being for very different

wavelengths. This is true for the other two channels as well, so we will only discuss relationships between DNB radiance and 4.05 μm and 12.01 μm brightness temperature.

Generally, unpopulated locations have a negative (red) correlation between DNB radiance and brightness temperature. While this relationship seems counterintuitive, it is likely due to lower lunar zenith angles during the colder winter months. Unlike the sun, the moon is higher in the sky during the winter months resulting in higher lunar irradiance. This relationship does not hold over land in the Qatar domain (Figure 13a and Figure 14a). This may be due to differences in land surface scattering or some other unexplained relationship and requires further study.

Most populated locations exhibit little-to-no correlation between DNB radiance and brightness temperature. This is most evident in the San Francisco Bay Area (Figure 13b and Figure 14b), Las Vegas (Figure 13c and Figure 14c), and St George (Figure 13d and Figure 14d) domains and indicates that the brightness of these locations is not dependent on physical temperature. In the Iraqi Oil Fields domain (Figure 13e and Figure 14e) strong positive (blue) relationships are seen surrounding oil wells indicating that DNB radiance is brighter for these locations when the flare from an oil well is hotter. Consequentially, it may be possible to constrain the brightness of the locations surrounding oil wells based on their physical temperature, however, due to the general instability of the light sources and the limited dataset for the domain it is difficult to draw more concrete conclusions.

The Qatar domain exhibits significantly different relationships between DNB radiance and infrared brightness temperature over land as compared with the other domains, even in populated locations. Some populated locations show positive (blue) correlation while other show negative (red). The positive relationships appear to coincide with locations of oil drilling and processing and have low stability (see Figure 3). The negative relationships, on the other hand, appear to coincide with dense areas of buildings. The relationships seen in the Qatar domain, in general, require additional research to be understood.

## 6 Study Limitations and Considerations

There are limitations to the methodology used in this study that should be considered.

1. As discussed above, the cloud screening methods used are problematic. The VCM is not able to capture all nighttime cloud and a subjective, expert-based final step is employed. This final step is "all-or-nothing" where scenes that are subjectively determined to be too contaminated are removed from the timeseries. Consequently, the timeseries we have constructed do still contain some cloud. Future efforts will be needed to improve nighttime cloud masking.

2. Use of Nearest Neighbour interpolation is a choice which may or may not be the best choice for future studies depending on the specific use case. Use of a higher order interpolation scheme (e.g. bilinear, cubic, etc.) would

certainly result in lower RSD in all cases but may also reduce our ability to constrain variability based on other known parameters. The choice of interpolation scheme is discussed in more detail in Appendix A.

3. Multiple studies have indicated the importance of changing surface albedo (due to snow in particular and vegetation index to a lesser degree) on the brightness of even bright city lights (Levin and Zhang, 2017; Román et al., 2015 and 2018). We have not considered the impacts of changing surface albedo in this study. For our particular study domains, most of which are desert, snow and changing vegetation index are minimal. The domain with the highest average annual snowfall is St. George, UT which receives 1.4 inches per year on average and has an average of 0.4 days per year with greater than 0.1 inches of snow (Arguez et al., 2010). Additionally, with the exception of the San Francisco domain, vegetation index is generally low throughout the year for all domains.

4. The data used for this study come from a sun-synchronous satellite. As such, each domain is seen at approximately the same time every night. This certainly reduces the variability that is able to be observed, but for retrievals based on DNB radiances from S-NPP VIIRS this is not really a limitation.

## 7 Summary and Conclusions

To develop a baseline understanding of the characteristics of nighttime light sources and aid efforts towards nighttime atmospheric retrievals of cloud and aerosol properties we have collected 18 months of data from the VIIRS Day/Night Band, four coincident brightness temperature channels, and several variables that describe Earth/satellite and Earth/Moon geometries. The data were cloud cleared using the VIIRS Cloud Mask (Sect. 3.1) and manual examination of the imagery (Sect. 3.2). We then examined the stability of a variety of terrestrial light sources as observed by the DNB in several different northern-hemisphere domains (Sect. 4). Additionally, we have explored the correlations that exist between DNB-observed visible radiance, infrared brightness temperature, and the geometric variables (Sect. 5).

Some general conclusions that can be drawn from this study include:

- Both populated and unpopulated locations exhibit a wide range of stabilities.
- Even the most stable locations in this study have some variability (on the order of 10-20% relative standard deviation).
- Some light sources are stable enough that they might provide the stability required to make accurate AOD retrievals (RSD < 0.2) but many others are not.
- Stable locations appear to be coincident with population centres such as the Financial District in San Francisco and the densely populated areas of Doha Qatar.
- Suburban locations vary over a wide range of RSD values, but can be relatively stable as well.
- Significant vertical structure can cause light to be blocked when viewed from the sides. These locations appear brighter when viewed from above than when viewed from the sides.

- Areas with a lack of vertical structure can exhibit the opposite correlation (i.e. brighter when viewed from the side).
- Some locations (e.g. airports and the Las Vegas "Strip") that might be expected to be useful as light sources for retrievals are unstable and would result in significant errors in retrievals that rely on those light sources.
- While the data in this study is limited for physically hot light sources, such as the flames emitted by oil wells, these light sources appear to be unstable are likely unreliable as light sources for retrievals.
- DNB radiance observed from unpopulated locations is generally unstable in time.
- Unpopulated areas near bright light sources are more stable and are brightest when the satellite is looking towards the light source due to forward scattering for the land/water surface.

Sect. 5 discusses potential sources for the variability observed in this study and examines the correlations between DNB radiance and other variables. We find that for unpopulated locations the lunar zenith angle, which correlates well with lunar phase for Suomi-NPP's 0130 local overpass time, likely explains much of the variability. However, proximity to visible emission sources, rugged terrain, and surface scattering effects have impacts on the brightness of unpopulated locations.

More to the point of this study, the brightness of populated locations exhibit a variety of different relationships. Due to the brightness of populated locations, which can be orders of magnitude higher in DNB radiance than unpopulated locations even under full moon conditions, populated locations have little dependence on lunar zenith angle and lunar phase (Sect. 5.1). The visible brightness of populated locations appears to correlate most strongly with satellite viewing geometry. As seen in Sect. 5.2, locations that contain a significant number of tall buildings appear brighter when viewed from above (low satellite zenith angle) than when viewed from oblique angles (high satellite zenith angle). Other locations that exhibit a similar negative correlation between satellite zenith angle and DNB radiance are locations with significant amounts of vertically pointing lights such as lighted highways and shopping complexes with large lighted parking lots. Conversely, less densely populated locations such as residential neighbourhoods appear brighter when viewed from an oblique angle (high satellite zenith angle), likely due to the directionality of light emission and a lack of occluding structures such as tall buildings. For locations that appear brighter when viewed from an oblique angle it is also important to consider which direction they are viewed from (Sect. 5.3). For populated locations, this directional dependence is likely caused by the direction of light emission from the source and occlusions such as tall buildings.

While most the observed brightness of most light sources shows little correlation to infrared brightness temperature (Sect. 4), temperature is an important factor for some physically hot light sources. For example, oil wells and refineries sometimes burn off excess gas resulting in visibly bright, physically hot light sources. As stated previously, the data for physically hot light sources is limited in this study, however, it can be said that these light sources exhibit a strong relationship between DNB radiance and infrared brightness temperature. With additional data and study it might be possible to constrain the brightness

of these physically hot emission sources using retrieved estimates of their physical temperatures, but that is a subject for further research.

The two major sources of error in this study are cloud contamination in our dataset and the use of nearest-neighbour interpolation to map our data to standard latitudes and longitudes. Despite our best efforts at cloud screening using both the automated VIIRS Cloud Mask and manual examination of the imagery, it is certain that some cloud contamination remains in the dataset. Improvement in this area would require development of a more robust nocturnal cloud mask, especially with regards to low-altitude cloud layers. Interpolation to standard latitudes and longitudes is required for development of timeseries, however, this process does not adequately account for the point-spread function of the sensor. More sophisticated methods that do not require interpolation of the data may yield better results.

Although this study has been limited in scope it indicates that different visible emission sources will have differing levels of usefulness for efforts to use terrestrial emissions as stable light sources due to their inherent temporal instability. The temporal stability of each light source should be examined prior to use in retrieval algorithms to avoid using unstable sources. Studies that endeavour to use terrestrial light sources in retrieval algorithms should also consider the impacts of viewing geometry on the observed brightness of light sources under cloud-clear conditions, even for the most temporally stable locations.

Further research will be required to develop datasets that can be used as the basis of retrieval algorithms. In the future we hope to expand this study to larger domains and over longer time periods using the data available from both Suomi-NPP and JPSS-1. Expanding the study to larger domains, however, will require improvements to cloud masking techniques to remove the need for expert screening. We are currently examining methods to improve upon the current IR-based cloud masks by including information from the DNB over light sources. Additionally, in future work will examine the stability and spatial characteristics of spatially grouped city lights to determine if stability and spatial variability over a larger area may prove to be more useful for retrieval algorithms. We also hope to extend the work we have done with correlation coefficients to develop models that are able to describe the variability of city lights under cloud-cleared conditions to reduce the uncertainty in assumed brightness.

## 8 Acknowledgements

This work was supported by the Office of Naval Research under Grant # N00014-16-1-2040, a project titled "Advancing Littoral Zone Aerosol Prediction via Holistic Studies in Regime-Dependent Flows." The data used for this work was collected from the Comprehensive Large Array-Data Stewardship System (CLASS) operated by the National Oceanographic and Atmospheric Administration (https://www.class.noaa.gov).

## Appendix A

Choice of interpolation scheme is an important consideration for this study. There are many interpolation schemes available, each of which have their own advantages and drawbacks. For this study we chose to use Nearest Neighbour (NN) interpolation but could have chosen a higher order scheme such as Bilinear (BL) interpolation if it better suited our use case. In this
appendix, we explore the impact on our data from NN and BL interpolation.

Nearest Neighbour interpolation is exactly what it sounds like. For each coordinate in a predefined set of output coordinates, the closest point in an input dataset is found. The value of that closest point is then assigned to the corresponding point in the set of output coordinates. There are consequences to choosing NN, especially in the case of pixels that contain a single bright
"point-source" that is significantly smaller than the instrument footprint. Take, for example, a case where there is a single, bright point-source of light whose brightness is stable in time and which has a normal emission function (i.e. it is the same brightness regardless of the viewing angle). Due to changes in viewing geometry, a satellite viewing that same point on successive nights may see it in any of several neighbouring footprints. This results in all of the light from the point source being assigned to one of any of those several neighbouring footprints and possible displacement of up to one whole footprint
from one observation to the next. When NN is used, this displacement is preserved in its entirety, so the point source would "jump around" between observations.

The impact of this can be seen in Figure 15 which shows DNB radiances on four different nights (a-d) in the same domain and the resulting RSD (e; note the colour scale differs here from the rest of the manuscript). The domain is centred at 30.15°N,
47.40°E and focuses on a small set of oil well gas flares in Iraq. Each of the four images uses NN. Since the gas flares are very small relative to the footprint size of VIIRS their observed locations shift from night to night. This results in high relative standard deviation (~2.0; Figure 15e) for the gas flares themselves.

Bilinear interpolation is slightly more complex than NN. For each coordinate in a predefined set of output coordinates, a set
of the four closest points is found in an input dataset. Weights are calculated based on the distance of the output coordinate from each of the four input points. The value at the output coordinate is the weighted sum of the values from the four input pixels. This process reduces the amount of movement in the interpolated imagery when compared to NN interpolation, but also results in data being smoothed over multiple pixels.

The images in Figure 16 use bilinear interpolation, but otherwise are the same as Figure 15. The point light sources from the gas flares are smoothed over multiple pixels and result in a decrease in movement of interpolated light source from night to night, but also results in an increase in the illuminated area. This results in reduced relative standard deviation (higher stability) for imagery produced using bilinear interpolation. Bilinear interpolation and other higher order interpolation schemes can

result in better looking imagery and is useful in many applications. However, we chose to use NN interpolation to preserve the variability that is observed by the DNB.

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

| Channel Name | Central Wavelength (μm) | Wavelength Range (μm) | Spatial Resolution at Nadir (m) |
|---|---|---|---|
| DNB | 0.700 | 0.50–0.70 | 742 (constant) |
| M13 | 4.05 | 3.97–4.13 | 750 |
| M14 | 8.55 | 8.4–8.7 | 750 |
| M15 | 10.76 | 10.26–11.26 | 750 |
| M16 | 12.01 | 11.54–12.49 | 750 |

**Table 1: Description of the VIIRS Channels used in this study including central wavelength, wavelength range, and spatial resolution. Note that, as described in Hilger et al., 2013, the "moderate" bands have three aggregation zones on each side of nadir to minimize the bowtie effect and the DNB has 32 aggregation zones on each side of nadir to produce a nearly constant resolution across the swath.**

| Location | Northwest Corner | Southeast Corner | Reason for Choice | Approx. 2015 Population | # Images |
|---|---|---|---|---|---|
| San Francisco, CA | 38.60 N, 123.10 W | 36.90 N, 120.90 W | Variable terrain; large population centre; frequent problematic low-cloud coverage; fishing boats | City: 866,320[1]<br>Metro: 4.657,985[2] | 204 |
| Las Vegas, NV | 37.00 N, 116.25 W | 35.25 N, 114.10 W | Large population centre; limited cloud contamination; significant light features (i.e. "The Strip") | City: 621,481[1]<br>Metro: 2,110,330[2] | 308 |
| St George, UT | 36.23 N, 114.67 W | 36.23 N, 112.51 W | Moderate population centre; limited cloud contamination; representative of rural town | City: 79,614[1]<br>Metro: 154,731[2] | 285 |
| Doha, Qatar | 26.11 N, 50.19 E | 24.39 N, 52.10 E | Large population centre; limited cloud contamination; significant oil and gas activity nearby; fishing boats | City: 633,000[3]<br>Country: 2,363,569[3]<br>*2018 Numbers* | 340 |
| Gas Flares, Iraq | 31.16 N, 46.30 E | 29.44 N, 46.31 E | Isolated oil wells with "flaring"; hot oil wells cause problems in VIIRS cloud mask | N/A | 95 |

**Table 2: Domains used in this study defined by their northwest and southeast corner coordinates in latitude and longitude. All domains are in stereographic projection and registered to 0.75 km spatial resolution (the nominal resolution of the DNB) on a 256x256 pixel grid. Summaries of the reasons for each domain choice are provided in the fourth column. The number of cloud-cleared, quality-controlled images used for each domain in this study is in the fifth column. Population estimate sources: 1) U.S. Census Bureau, Population Division; Annual Estimates of Resident Population for Incorporated Places of 50,000 or More: April 1 2010 to July 1 2017, 2) U.S. Census Bureau, Population Division; Annual Estimates of the Resident Population; April 1 2010 to July 1 2017, 3) CIA World Factbook.**

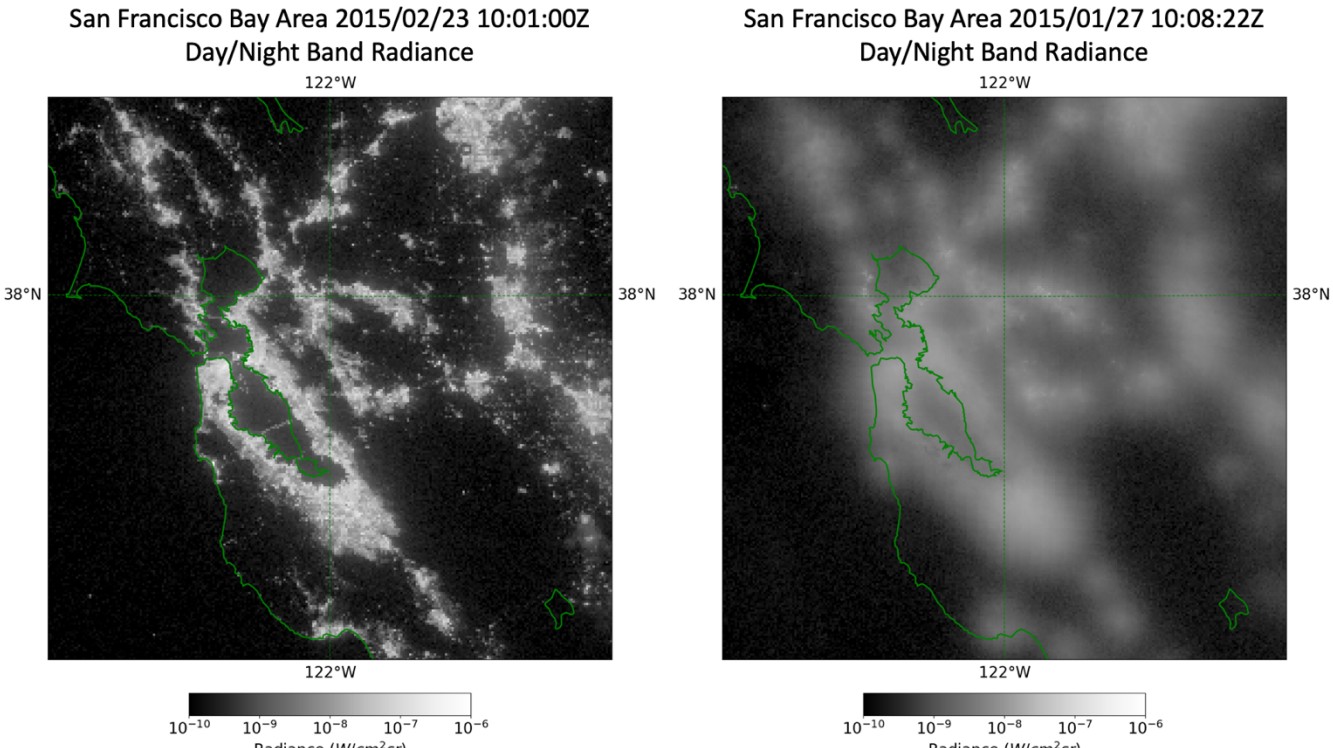

**Figure 1: VIIRS Day/Night Band radiances over the San Francisco Bay Area (left) under clear conditions and (right) under cloudy conditions. Under clear conditions structure is visible within the terrestrial lights showing roads, bridges, areas of high and low population density, etc. Under cloudy conditions, however, only the general shape of cities can be observed. This illustrates the sensitivity of observed radiances to intervening cloud layers and the importance of appropriate cloud screening in this study.**

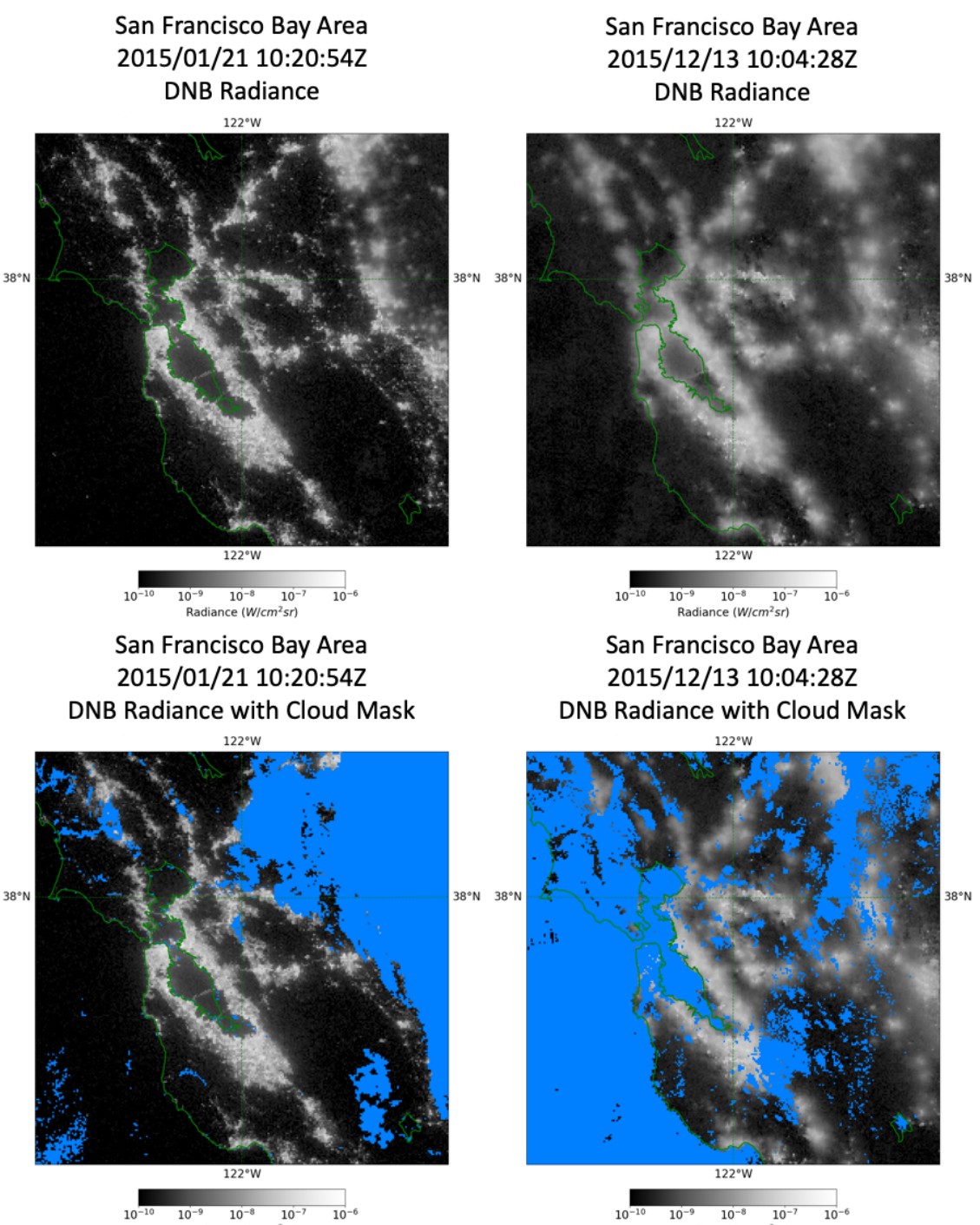

**Figure 2: VIIRS Day/Night Band radiances over the San Francisco Bay Area without a cloud mask applied (top) and with clouds identified by the VIIRS Cloud Mask coloured in blue (bottom). (left) A situation where the cloud mask performs well. (right) A situation where the cloud mask misses some cloud over land as inferred by the blurring of the underlying surface light structure.**

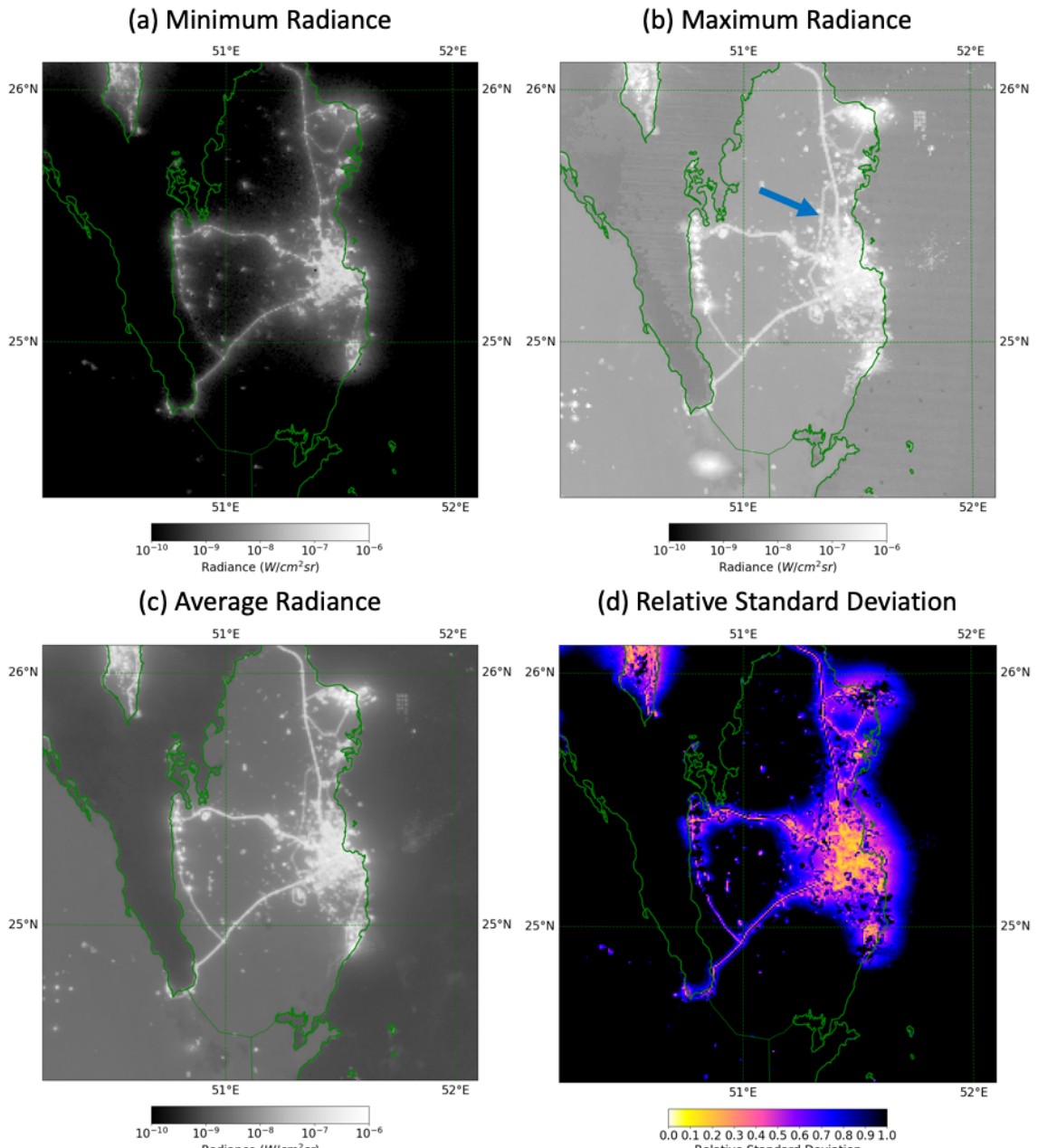

**Figure 3: Statistics for the Qatar domain based on observations from January 1 2015 through June 30 2016 showing spatially resolved (a) minimum, (b) maximum, (c) average, and (d) relative standard deviation of cloud-cleared DNB radiances.**

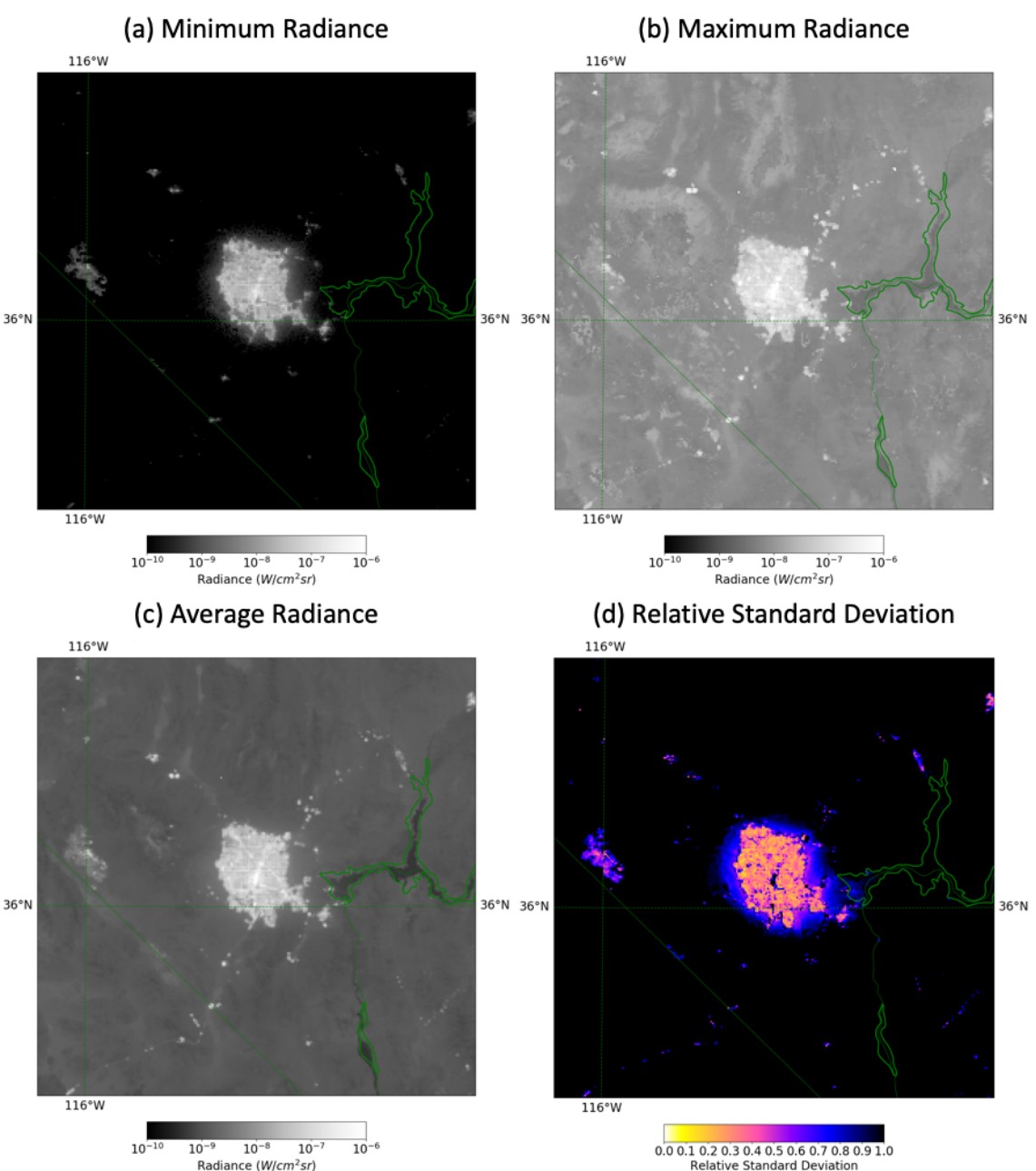

**Figure 4: Same as Figure 3 but for Las Vegas, NV, USA.**

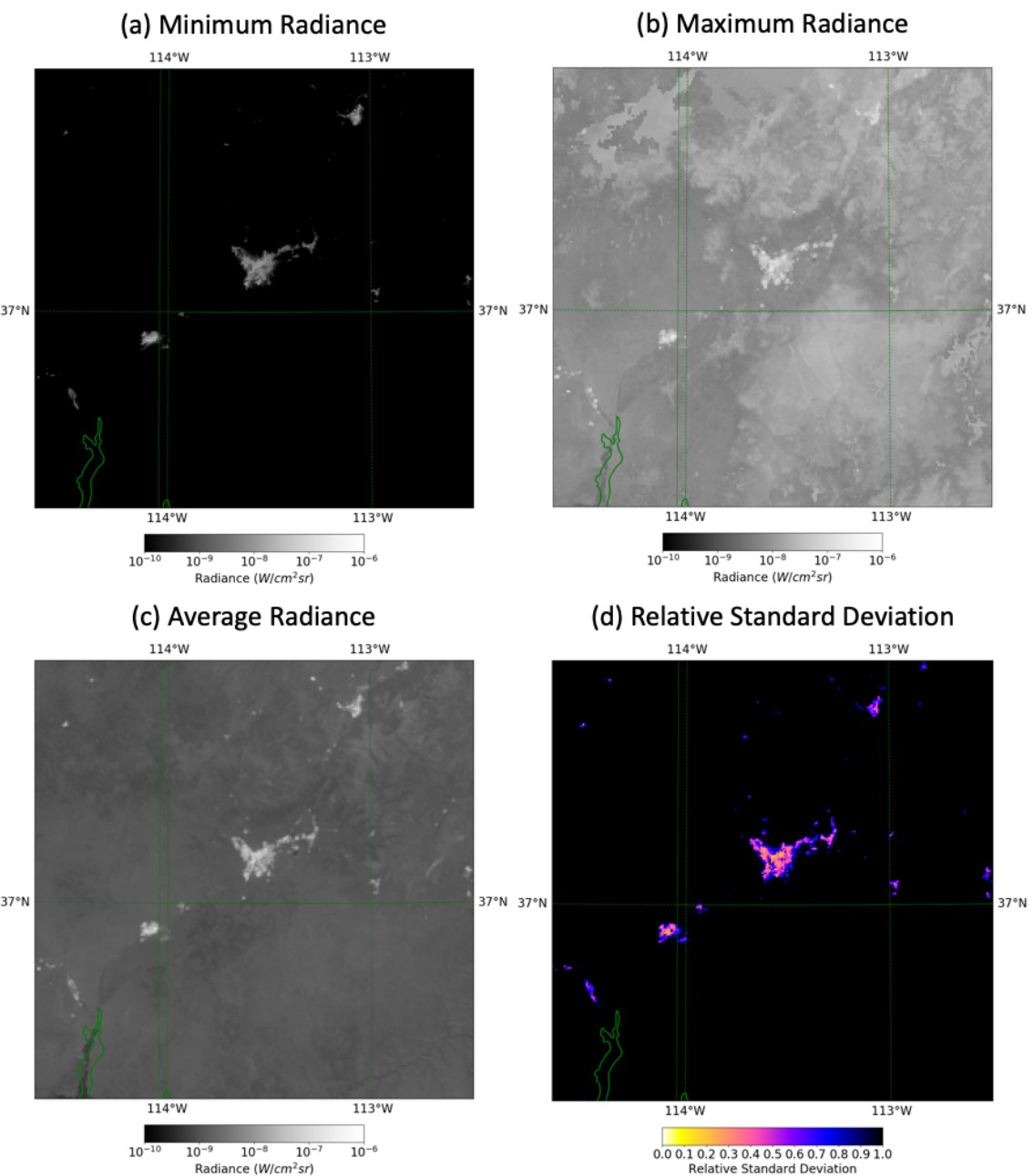

**Figure 5: Same as Figure 3 but for St. George, UT, USA.**

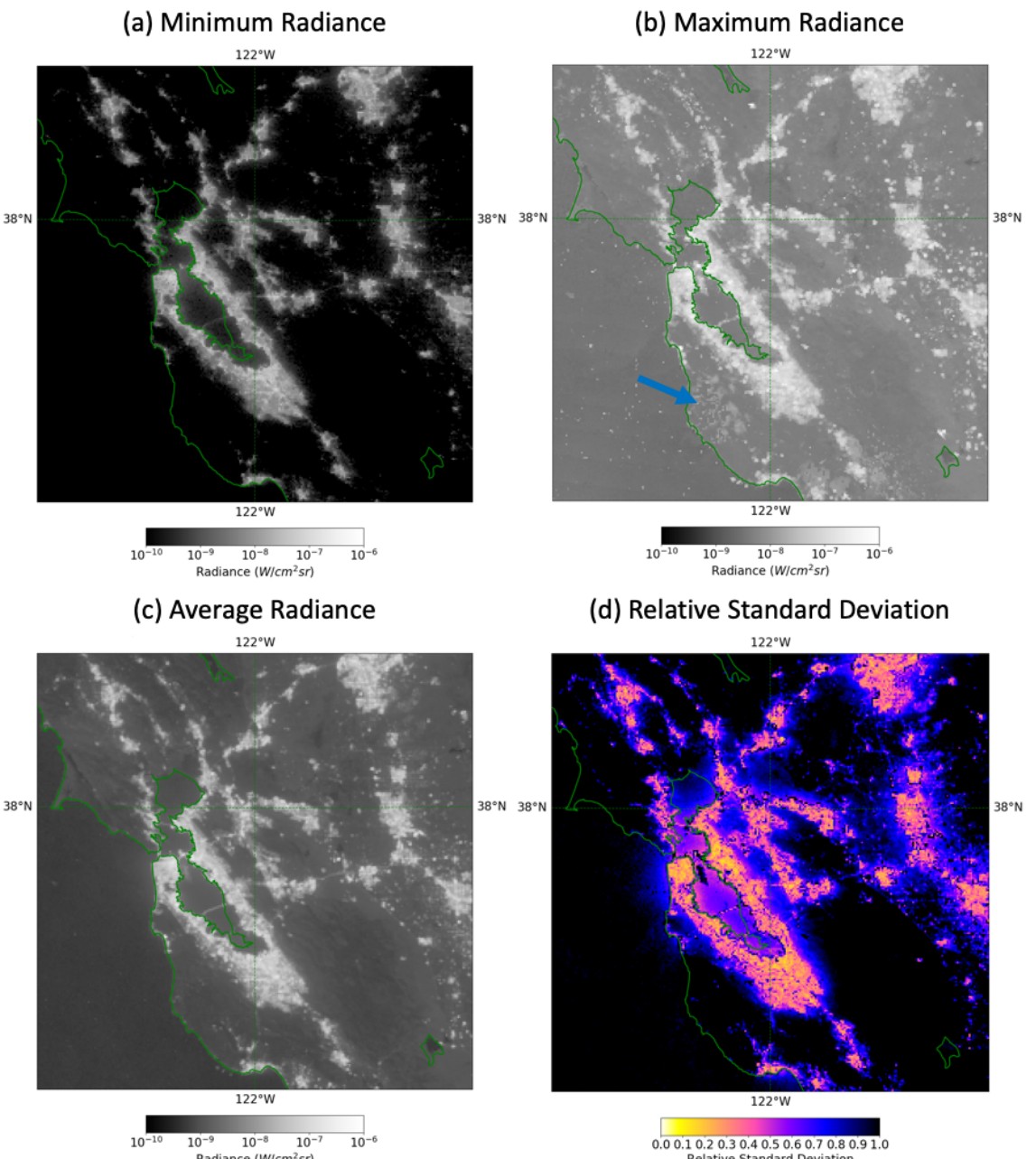

**Figure 6: Same as Figure 3 but for the San Francisco Bay Area, California, USA.**

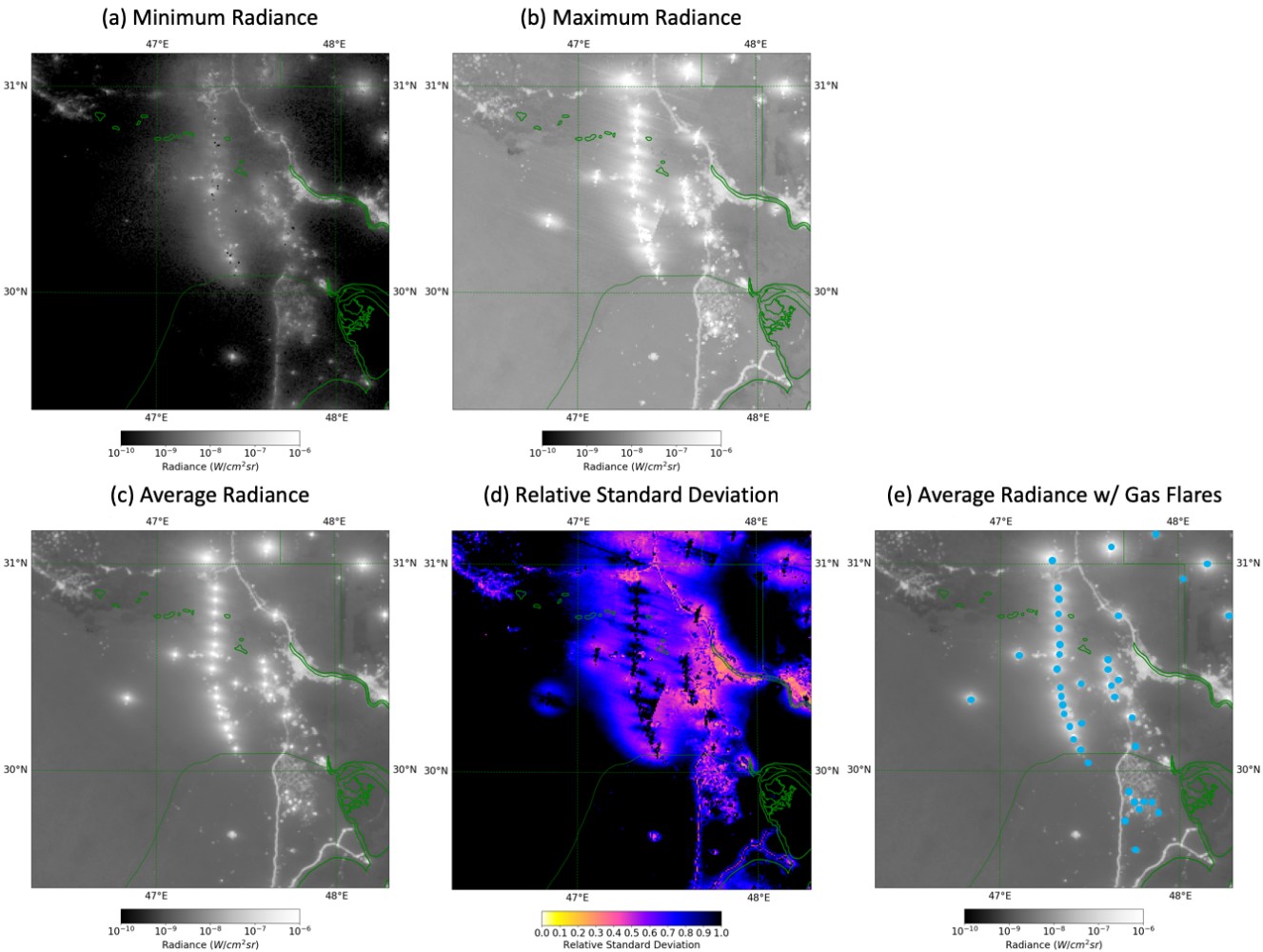

**Figure 7: (a-d) Same as Figure 3 but for oil fields in southwestern Iraq, approximately 50 km west of Basra. (e) Average DNB radiance with gas flares marked in blue.**

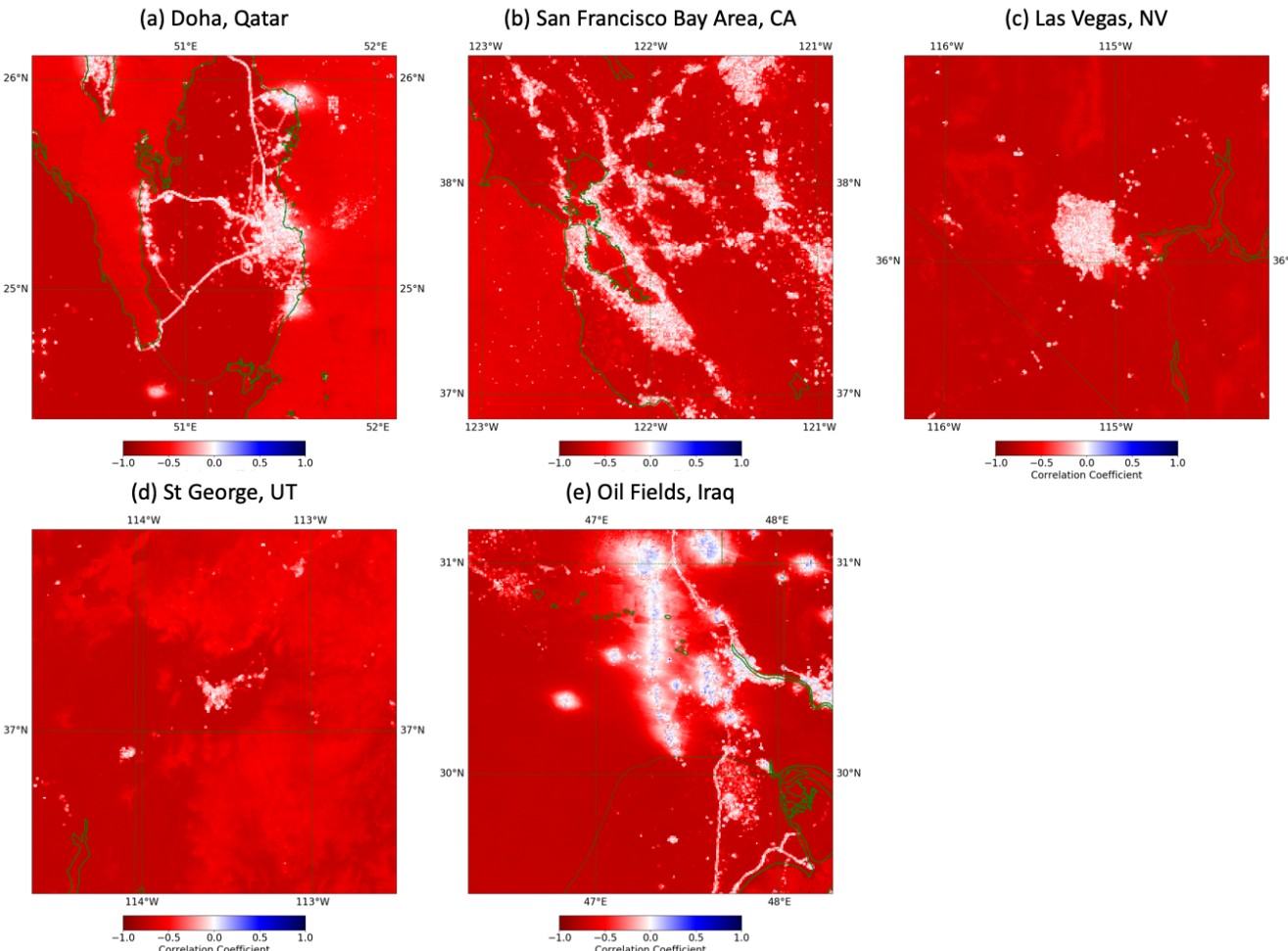

**Figure 8: Spatially-resolved correlation coefficient between DNB radiance and Lunar Zenith angle (0° to 180°) for: (a) Qatar; (b) San Francisco Bay Area, CA; (c) Las Vegas, NV; (d) St. George, UT; (e) Iraqi Oil Fields. Positive (blue) values indicate that the location is brighter for higher lunar zenith angles (i.e., when the moon is lower towards the horizon). Negative (red) values indicate that the location is brighter for lower lunar zenith angle (i.e., when the moon is more overhead). Values of zero (white) indicate that there is no correlation between DNB radiance and lunar zenith angle.**

## Correlation Coefficient: DNB Radiance vs Satellite Zenith Angle

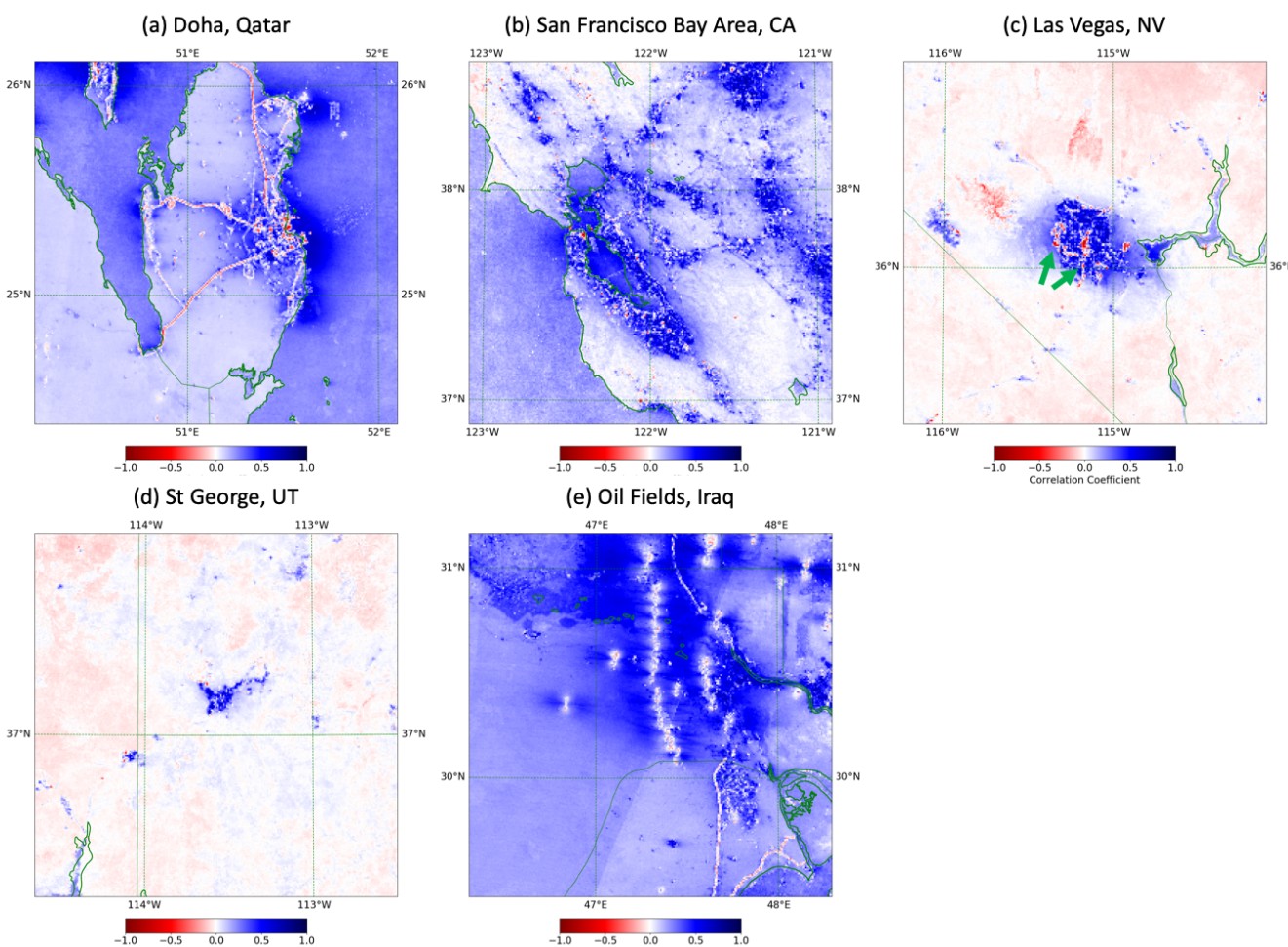

**Figure 9: Same as Figure 8 but for correlations between DNB radiance and Satellite Zenith angle (0° to 70°). Positive (blue) values indicate that a location appears brighter for higher satellite zenith angles (i.e. when the location is viewed obliquely). Negative (red) values indicate that a location appears brighter for lower satellite zenith angle (i.e. when viewed from above). Values of zero (white) indicate that there is no correlation between the brightness of the location and viewing angle.**

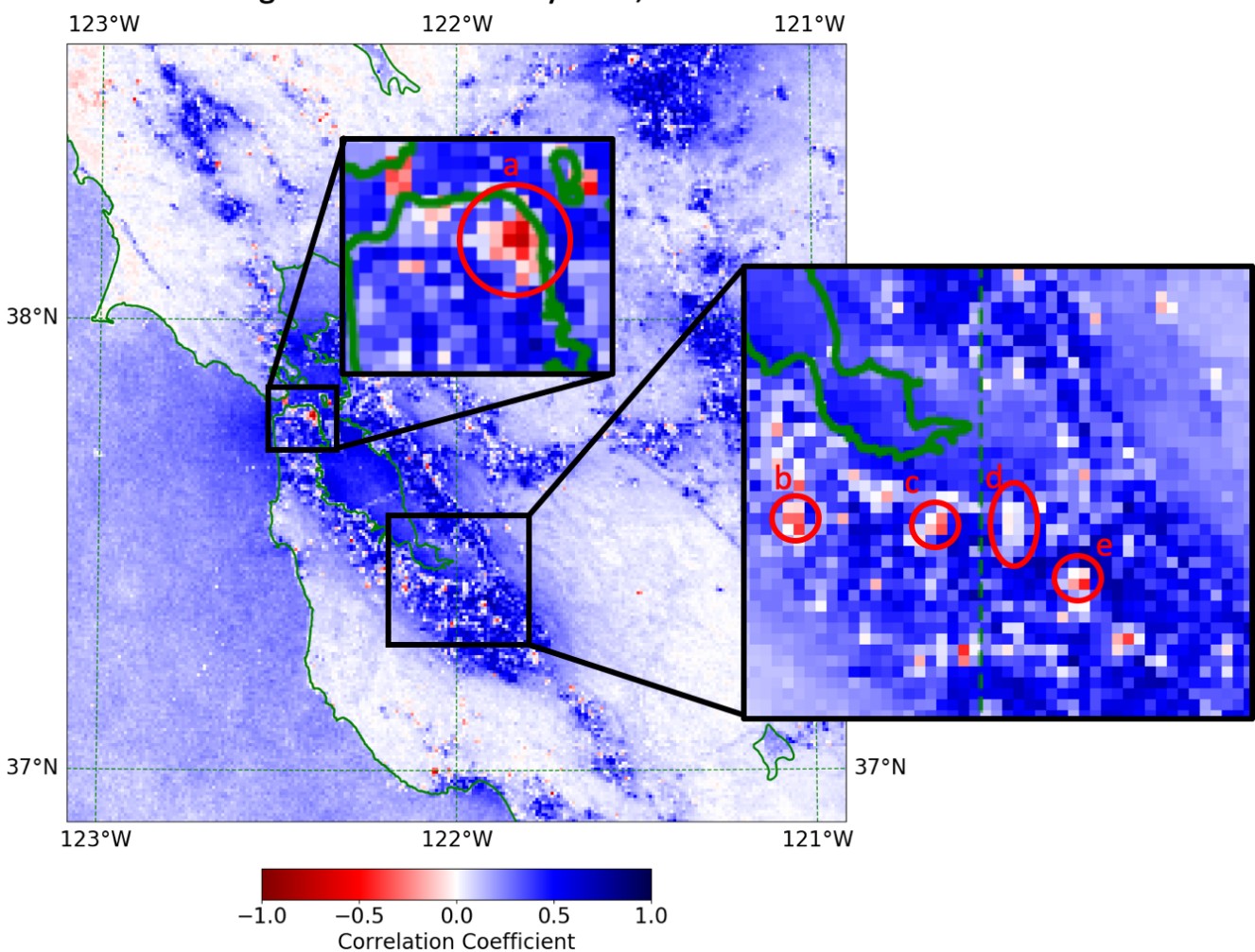

**Figure 10: Correlation coefficient between DNB radiance and Satellite Zenith angle for the San Francisco Bay Area with two zoomed regions over (left) San Francisco's Financial District and (right) the San Jose/Santa Clara area. In the left-hand zoom, San Francisco's Financial District (circled and labelled "a") has a strong negative correlation, indicating that it is brighter when viewed from above. In the right-hand zoom, the four red circles correspond to (b, c) areas with large numbers of shopping centres, (d) San Jose International Airport, and (e) downtown San Jose.**

## Correlation Coefficient: DNB Radiance vs Directional Satellite Zenith Angle

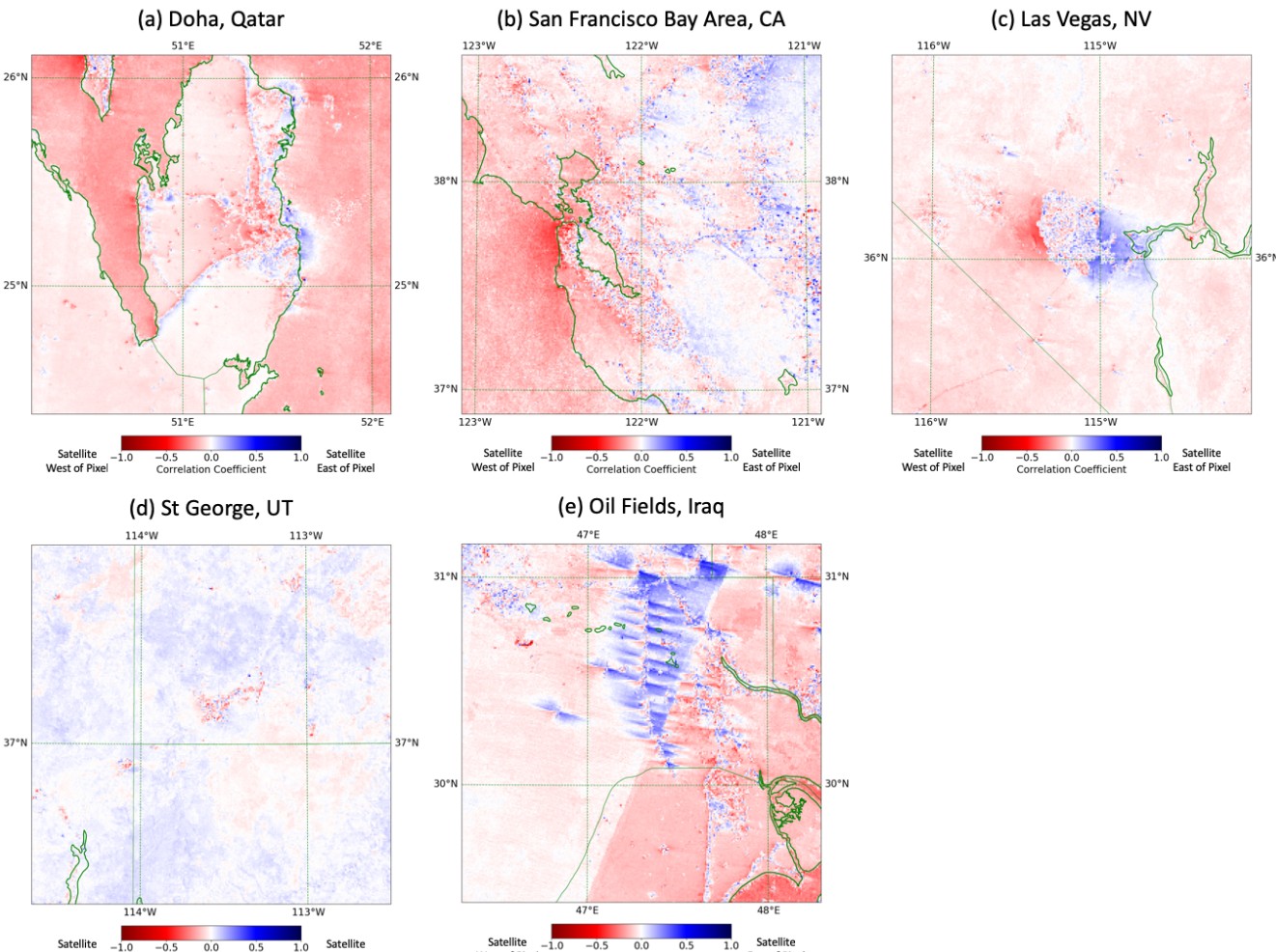

**Figure 11: Same as Figure 8 but for DNB radiance vs Directional Satellite Zenith angle (-70° to 70°). Positive (negative) values of Directional Satellite Zenith angle indicate that the satellite is positioned to the east (west) of the location that it is viewing. Positive (blue) values of correlation coefficient indicate that a location appears brighter when viewed by a satellite positioned to the east of that location (i.e. when the DNB is looking toward the west). Negative (red) values indicate that a location appears brighter when viewed by a satellite positioned to the west of that location (i.e. when the DNB is looking toward the east). Zero correlation (white) indicates that a location's brightness is independent of viewing direction.**

## Correlation Coefficient: DNB Radiance vs Directional Satellite Zenith Angle
## Moonless Nights

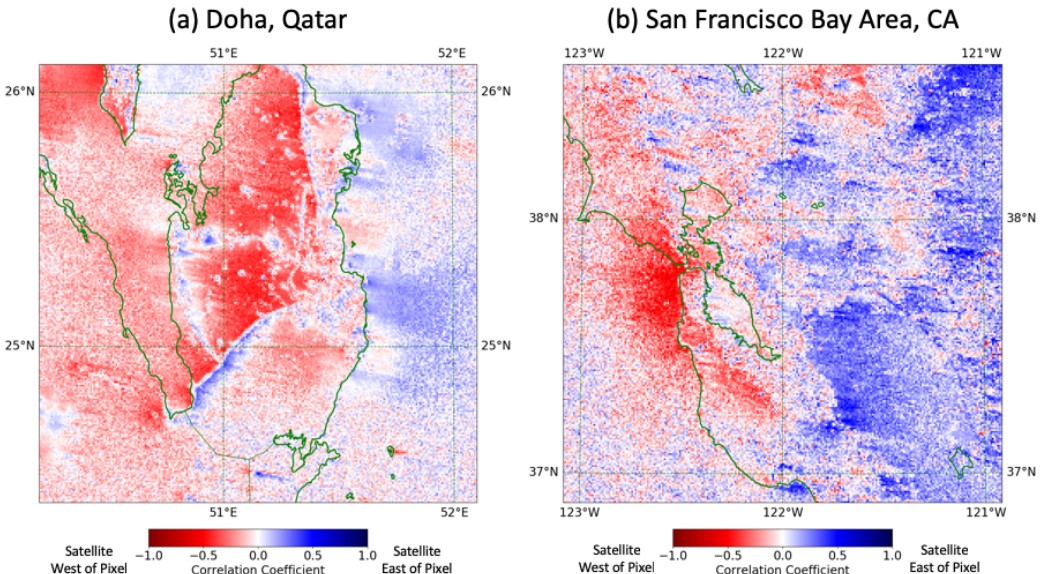

**Figure 12: Correlation coefficient on moonless nights (lunar zenith angle > 95°) for DNB radiance with Directional Satellite Zenith angle (-70° to 70°) for (a) Qatar and (b) the San Francisco Bay Area, CA. Negative (red) values indicate that a location appears brighter when viewed from the west (i.e. DNB looking toward the east). Positive (blue) values indicate that a location appears brighter when viewed from the east (i.e. DNB looking toward the west). Values of zero (white) indicate that there is no correlation between the brightness of the pixel and viewing direction.**

## Correlation Coefficient: DNB Radiance vs 4.05μm Brightness Temperature

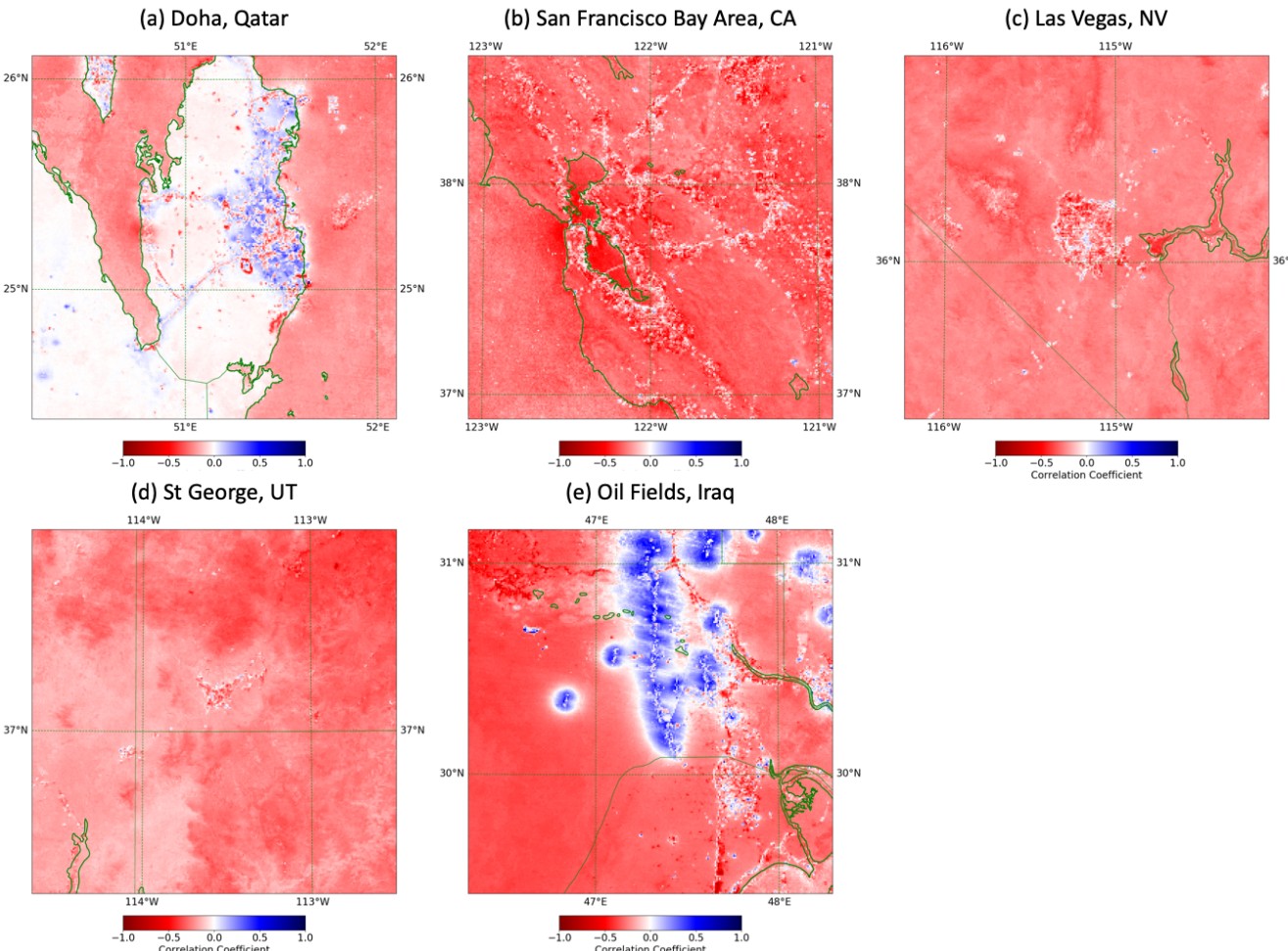

**Figure 13: Same as Figure 8 but for correlation between DNB radiance and 4.05 μm brightness temperature.**

# Correlation Coefficient: DNB Radiance vs 12.01µm Brightness Temperature

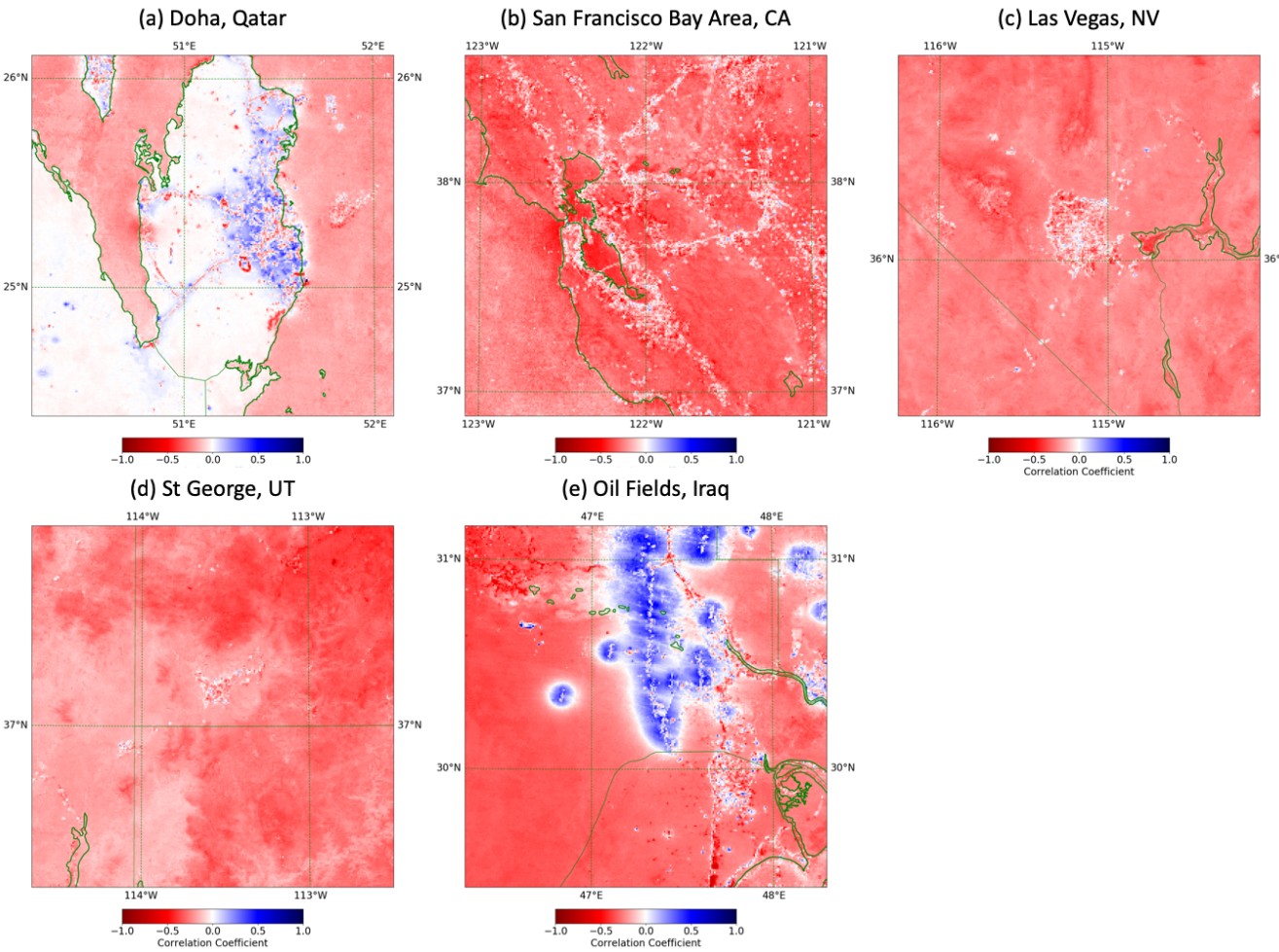

**Figure 14: Same as Figure 8 but for 12.01 µm brightness temperature.**

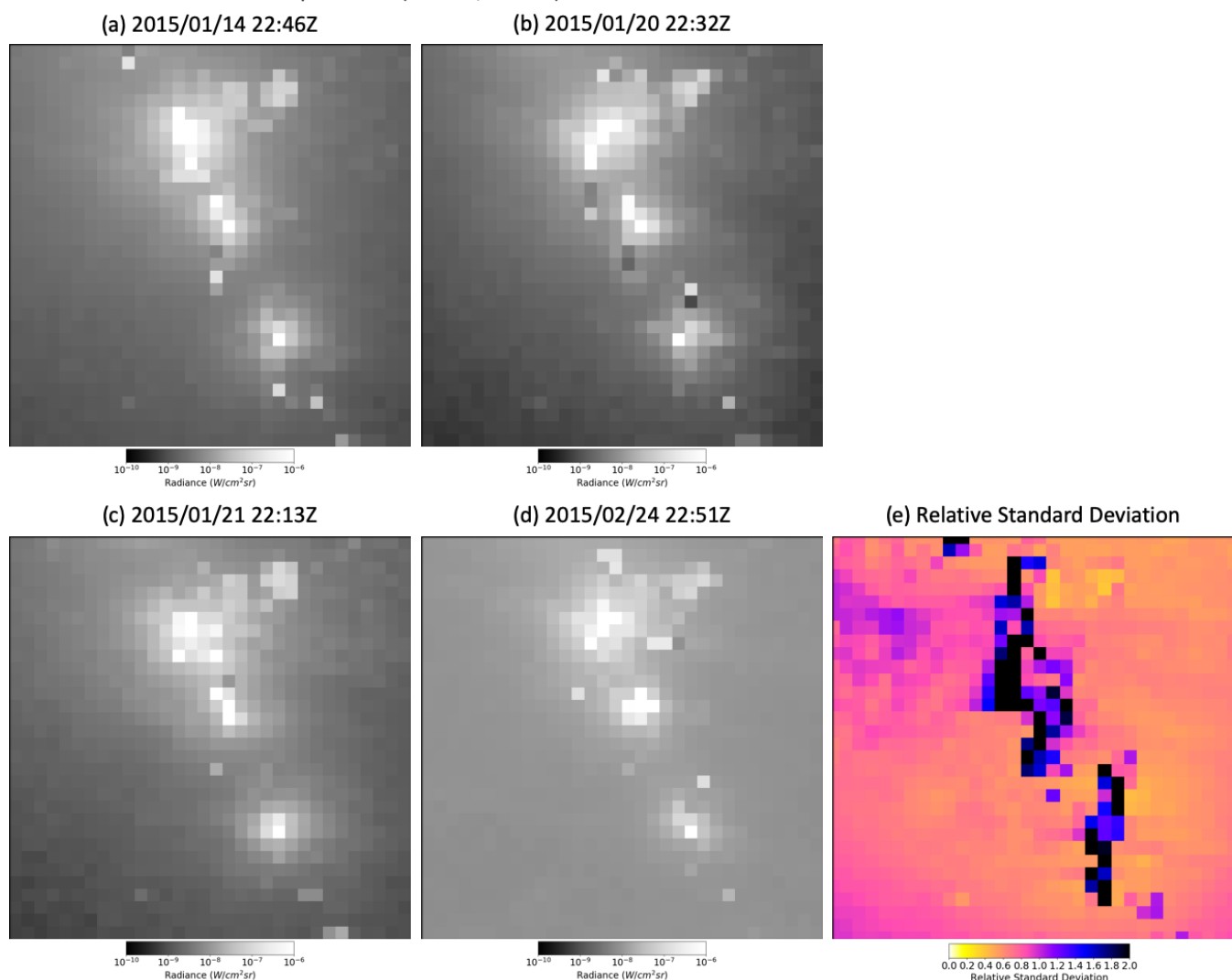

**Figure 15: Examples showing the use of Nearest Neighbour interpolation. The DNB radiance images (a-d) depict "point-source" lights produced by oil wells in Iraq. Note that, from night to night, individual light sources appear to move within their neighbourhood of pixels. This interpolation-caused movement causes relative standard deviation (e) to be low for point light sources.**

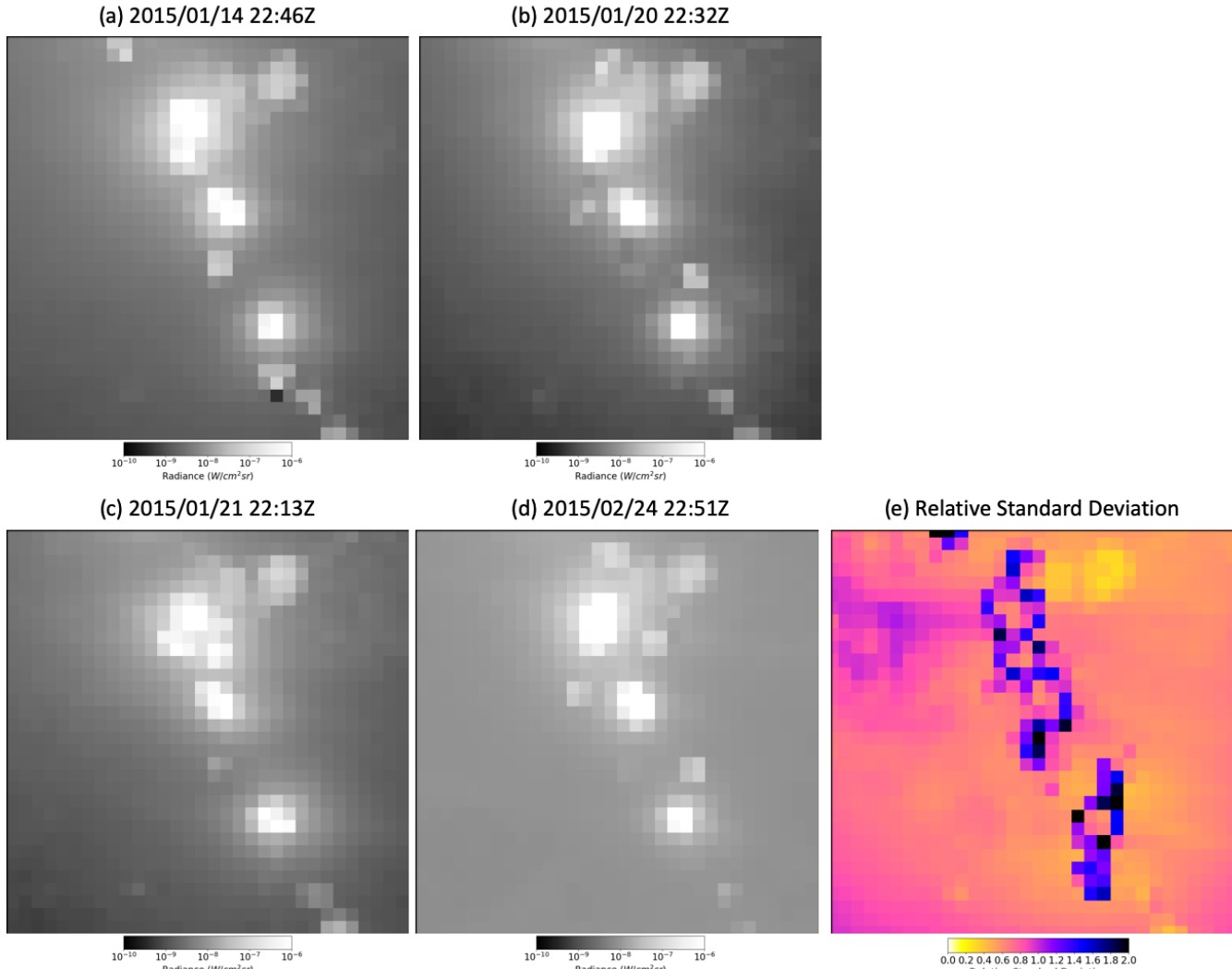

**Figure 16: Same as Figure 15 but using Bilinear interpolation. Note that individual "point-sources" move relatively little from night to night but are distributed across multiple pixels. The relative standard deviation (e) shows relatively high stability, but this is a consequence of spreading point sources over multiple pixels.**