# Peer review of "Assessing the Stability of Surface Lights for use in Retrievals of Nocturnal Atmospheric Parameters"

_Atmospheric Measurement Techniques, 2019_

## Referee Comment (RC2) · Anonymous Referee #2 · 26 Jun 2019

Dear Authors, First let me thank you for a well-written manuscript on this interesting study. The topic of quantitative characterization of nighttime environments is an exciting area, where scientists are rediscovering fundamental consequences of radiative transfer in novel contexts.

I think this study is sound and publishable, even though it is somewhat preliminary and reads largely as an exploratory data analysis. It would benefit from a bulletized summary of the conclusions, and a further summary of priority areas for further research.

I have arranged my comments roughly from most to least scientific importance, with typographic errors at the bottom. Good luck in preparing revisions, and thank you for your

hard work on this study. I appreciate your acknowledgement of the errors introduced by the nearest-neighbour interpolation used to generate your time series. However, you should specifically consider the impact of this resampling on the scenes which contain point sources. These point sources will not reliably be contained in the same pixel using your resampling approach, and the high variability and poor correlations over Iraq are likely to be primarily a function of this spatial error. Examination of isolated point sources such as gas wells is definitely a place where a more sophisticated approach to spatial data in required.

In your discussion of the maximum and minimum (and other stats) constructed scenes, you should clarify your expectations from these scenes. For pixels containing surface light sources, the brightness maximum should be the clearest night in the time series. For darker pixels adjacent to those sources, the brightness maximum should be the maximum atmospheric scattering (longest atmospheric path). This is consistent with what is seen in the patterns of correlation you found between satellite zenith and DNB, and should be discussed at the outset.

Page 11 line 15 end of Section 4. It's puzzling to say that DNB-IR correlation is beyond the scope of your study when the next section of the paper is about DNB-IR correlation. Since you are discussing the limitations of your study with regard to gas flares, this is a good place to mention the role of spatial resampling error in studying these areas.

I think it is worth noting that satellite aerosol retrieval algorithms going all the way back to Kaufman 1997 have used the variance in the visible brightness as a means of detecting and screening residual and subpixel cloud. This emphasizes both the opportunity and the challenge of using brightness variations as diagnostic of atmospheric conditions.

Page 17 line 29: I think the finding that the DNB radiance time series show the signal of forward scattering of anthropogenic light sources by land surfaces is a significant one and should be revisited here. Abstract-first sentence- this sentence is clumsy

and perhaps ungrammatical. 'Sensitivity' is 'information typically provided by visible spectrum observations'?

Page 12 Line 1 Start of Section 5. I think you can make this shorter by simply stating "The results below are shown for all sites including Iraq, but statistical strength for the Iraq domain is weaker due to smaller sample size for the reasons discussed above."

Page 13 Line 24 'While this is discussed further in Sect. 5.3, suffice to say...' Maybe just say "This effect is discussed in Section 5.3"

Page 15 line 19 "correlation...noisy" I think a better word is 'weak' when describing a correlation.

Page 15 line 21 "the same bias" do you mean the same trend?

Page 15 line 33 "DNB radiance and four brightness temperature from four" -> "DNB radiance and brightness temperature from four"

---

## Author Comment (AC1) · 21 Sep 2019

Dear Anonymous Reviewer,

Thank you for taking the time to review our manuscript and for your constructive comments and suggestions. You raise some good points that needed to be addressed. We appreciate the time you took to provide this thorough evaluation of our methods and their weaknesses. Please see our responses to your three major areas for improvement below.

1.  Throughout the paper, it is unclear what is the basic requirement for a high-quality nighttime aerosol retrieval using the available sample of stable point source data.

    a.  While this work is described by co-authors (J. Zhang), it is still unclear what is the minimum required sample to establish an aerosol retrieval for each specific study domain. Would a single pixel-based source be sufficient to estimate nighttime AOD over the regions of interest used (256 x 256 pixels)? Are more stable points needed? If so, what would be needed, sampling-wise, in order to ensure a routine retrieval of aerosols using this method?

        **Response:**
        We have added a new paragraph on Page 3 describing the requirements of aerosol retrieval algorithms that rely on city lights as their source of visible light. The paragraph also clarifies that the results of this study are not intended to be specific to one algorithm and should be useful for any algorithm that uses city lights as their source of visible light. The new paragraph reads:

        "This analysis provides a first step towards characterization of anthropogenic light sources for use in retrievals and is intended to be agnostic of the retrieval algorithm. Some algorithms may be capable of performing retrievals using attenuation of light emitted by isolated point sources. Others may rely on the amount of "blooming" observed around a light source to retrieve optical depth. It may also be possible to retrieve optical depth by observing changes in the brightness or spatial structure (e.g. spatial variance) of groups of well-characterized light sources. Regardless of the algorithm employed, understanding the variability in anthropogenic light emissions and its causes is important to the problem of retrieving optical depth at night."

    b.  Then, can you comment on how the methodology be scaled up at the region-level and over a sufficiently large global sample (urban-lit areas, only comprise less than 3% of the Earth's land mass). The authors should provide some commentary (and background/statistics/sampling estimates), which would then put the results of this paper into perspective.

        **Response:**
        Estimates of urban extent vary widely (Schneider et al., 2009). Given the estimate range, it might be reasonable to estimate that total urban extent is approximately 0.5% of Earth's land area (about 700,000 km$^2$). If all of the urban area were useable in optical

depth retrievals, a single VIIRS instrument, whose spatial resolution is 750 m, would be able to make about 1.2 million observations per night.  If only 1% of the urban-lit area is usable the number would drop to about 12,000 observations per night.  How much of the global urban area is useful for optical depth retrievals will be algorithm and application dependent as well as dependent on how well the brightness of each light source can be constrained.

2. The authors seem often confound basic terminology of reflectance/and nighttime remote sensing nomenclature, particularly when it comes to surface-related phenomena.  In most cases, what they deem "Lunar Geometry" or "Surface Scattering" is formally defined as "Lunar-reflectance", Lunar BRDF, as well as surface albedo and reflectance anisotropy.  That is, the authors described the surface phenomena as if it is only influenced by lunar phase and satellite/source view illumination angle of capture, when in reality there are additional factors that are intrinsic to surface conditions, which are virtually ignored.

This lack of understanding stems from the author's push towards characterizing stable point sources of high-intensity radiances >20nW, which are well-above the magnitude of lunar variation (< 10nW).  However, previous studies have documented how surface phenomena is in fact an influencing factor of stability of stable point sources.  For instance:

N. Levin, Q. Zhang, A global analysis of factors controlling VIIRS nighttime light levels from densely populated areas Remote Sens. Environ., 190 (2017), pp. 366-382, 10.1016/j.rse.2017.01.006

M.O. Román, E.C. Stokes Holidays in lights: tracking cultural patterns in demand for energy services Earth's Futur., 3 (2015), pp. 182-205, 10.1002/2014EF000285

Román, M.O.; Wang, Z.; Sun, Q.; Kalb, V.; Miller, S.D.; Molthan, A.; Schultz, L.; Belle, J.; Stokes, E.C.; Pandey, B.; et al. NASA's Black Marble nighttime lights product suite. Remote Sens. Environ. 2018, 210, 113–143.

M.M. Bennett, L.C. Smith: Advances in using multitemporal night-time lights satellite imagery to detect, estimate, and monitor socioeconomic dynamics, Remote Sens. Environ., 192 (2017), pp. 176-197, 10.1016/j.rse.2017.01.005

The most obvious effect not being considered is the influence of snow cover, which will most likely affect the RSD of Study Domains in temperate regions, given the 18-month sample size (Particularly St George UT).  See for example Figure B1 in (Roman and Stokes, 2015), which shows how the presence of snow cover influences the stability of point sources for large US cities.

**Response:**
This is a good point that will need to be explored as this type of analysis is employed over larger regions.  At least with respect to snow and NDVI, however, this is a relatively small

concern for the domains chosen.  None of the study domains have significant snowfall in an average year.  The study domain with the highest average annual snowfall is St George, UT which receives 1.4 inches per year and an average of 0.4 days per year with snowfall greater than 0.1 inches.  Additionally, most of the study domains are desert regions and have little annual cycle in NDVI.  The lone exception, San Francisco, does have an annual cycle in NDVI that may need to be considered in future studies.

We have added Section 7 which discusses some of the limitations of this study including this limitation.

3.  The authors seem to make conclusions about RSD and its dependency to urban built environments based on simple inferences collected from manual examination of specific areas within cities (e.g., large/tall building located in the San Francisco region, residential sectors, large roads, etc., as noted in page 13, line 32.) The analysis needs to be strengthened by a more quantitative assessment based on a control variable that characterizes the urban built area (e.g., % Urban cover). A dependency between urban density and RSD should be explored in this paper to make a more definitive conclusion.

The datasets are already available to come up with such an assessment. See for example, work by:

T. Esch, W. Heldens, A. Hirne, M. Keil, M. Marconcini, A. Roth, J. Zeidler, S. Dech, E. Strano Breaking New Ground in Mapping Human Settlements From Space The Global Urban Footprint (2017)

T. Esch, M. Marconcini, A. Felbier, A. Roth, W. Heldens, M.Huber, M. Schwinger, H. Taubenbock, A. Muller, S. DechUrban footprint processor-fully automated processing chain generating settlement masks from global data of the TanDEM-X mission, IEEE Geosci. Remote Sens. Lett., 10 (2013), pp. 1617-1621, 10.1109/LGRS.2013.2272953

Also see Figure 12a in, Roman et al., 2018:
https://www.sciencedirect.com/science/article/pii/S003442571830110X#bb0125

**Response:**
This is a good point that should be considered.  We intend to explore this issue in future work.  We have added language to the conclusions to discuss future work on this and other issues.

---

## Author Comment (AC3) · 21 Sep 2019

Dear Anonymous Reviewer,

We would like to thank you for taking the time to review our manuscript and for your constructive comments and suggestions. We have addressed each of the points raised both in the text and below. Please note the addition of Appendix A which discusses our choice to use Nearest Neighbour interpolation rather than a higher order interpolation scheme like Bilinear interpolation.

… would benefit from a bulletized summary of the conclusions…

> We have bulletized some of the main conclusions, but left some of the discussion to provide more details.

… further summary of priority areas for further research.

> We have expanded this section slightly and placed some emphasis on the need for improve cloud masks at night. Moving forward with this research will require better cloud screening that does not need a person to assess each individual image.

In your discussion of the maximum and minimum (and other stats) constructed scenes, you should clarify your expectations from these scenes. For pixels containing surface light sources, the brightness maximum should be the clearest night in the timeseries. For darker pixels adjacent to those sources, the brightness maximum should be the maximum atmospheric scattering (longest atmospheric path). This is consistent with what is seen in the patterns of correlation you found between satellite zenith and DNB and should be discussed at the outset.

> We added three new paragraphs to the Section 4 introduction describing what can be expected from the minimum, maximum, and average radiance images. Also, the paragraph that describes Relative Standard Deviation has been extended to describe what can be expected from the RSD images.

Page 11 line 15 end of Section 4. It's puzzling that to say that DNB-IR correlation is beyond the scope of your study when the next section of the paper is about DNB-IR correlation. Since you are discussing the limitations of your study with regard to gas flares, it is a good place to mention the role of spatial resampling error in studying these areas.

> We edited this section to remove the point about DNB-IR correlation being beyond the scope of the study.

> Appendix A has been added to discuss the impacts of using Nearest Neighbour interpolation vs Bilinear interpolation.

I think it is worth noting that satellite aerosol retrieval algorithms going all the way back to Kaufman 1997 have used the variance in visible brightness as a means of detecting and

screening residual and subpixel cloud. This emphasizes both the opportunity and challenge of using brightness variations as a diagnostic of atmospheric conditions.

> Edited the first paragraph of section 3.3 to include the citation that describes this process for the MODIS aerosol algorithms.

> "The variance in visible radiance has been used for nearly two decades as a method of screening residual cloud. For example, Martins et al. (2002) expand upon the MODIS cloud mask for use in aerosol applications by masking regions of high spatial variance in visible brightness as cloud. Conversely, using expert analysis, we identified any overpasses containing significant reductions in spatial variance of the terrestrial light sources (e.g. **Error! Reference source not found.**, right), indicating the presence of cloud or aerosol that was not masked by the VCM."

Abstract-first sentence- this sentence is clumsy and perhaps ungrammatical. 'Sensitivity' is 'information provided by visible spectrum observations'?

> Edited to read "Detection and characterization of aerosols is inherently limited at night because the important information provided by visible spectrum observations is not available and infrared bands have limited sensitivity to aerosols."

Page 17 line 29: I think the finding that the DNB radiance time series show the signal of forward scattering of anthropogenic light sources by land surfaces is significant and should be revisited here.

> Thank you for pointing this out. Leaving this out of the conclusions was an oversight and it has been added.

Page 13 Line 24 'While this is discussed further in Sect. 5.3, suffice to say…' Maybe just say "This effect is discussed in Section 5.3"

> This has been reworded

Page 15 line 19 "correlation…noisy" I think a better word is 'weak' when describing a correlation.

> Agreed, corrected

Page 15 line 21 "the same bias" do you mean the same trend?

> Yes, "trend" is the appropriate word here. Corrected.

Page 15 line 33 "DNB radiance and four brightness temperature from four" -> "DNB radiance and brightness temperature from four"

Corrected

Again, thank you for taking the time to review our manuscript.

---

## Author Response (AR2)

[revised manuscript text omitted]

Hi Anthony,

Thank you for the constructive edits.  I have made all of the suggested edits.  They are included in the track changes for the uploaded document.  See below for comments on the various changes.

Sincerely,
Jeremy Solbrig

Pg. 2, line 23: The year on the Walther et al. reference is given as 2013 in the text, but is listed as 2018 in the reference list. Please make consistent to the proper year.

**Updated to 2018 in the text.**

Pg. 3, lines 6-12 and lines 14-21: These two paragraphs seem to be close repeats of each other. Please consolidate into one paragraph.

**Consolidated the two paragraphs into one.  The new paragraph starts on line 6.**

Pg 4, line 30: "Geolocation data were gathered" did this study use terrain-corrected geolocation data or not? Please clarify.

**Added notation that the geolocation data were terrain corrected on line 24 (location changed due to previous edit).**

Pg. 7, lines 13-14: The sentence "The variance in visible radiance…" is a close repeat of the sentence in lines 7-8. Please only put in one place.

**Removed the sentence that started at line 13.**

Pg. 7, line 19: Suggest changing the sentence to read: 'Note that, while this method substantially helps screen cloudy scenes,…"

**Agreed, updated.**

Pg. 8 line 27: "While simplistic in construct, these statistics" change to "These simple statistics" or even "These statistics"

**Updated to read "These statistics".**

Pg. 13, line 30: Use the American 'artifact'

**Updated to "artifact".**

Pg. 19, line 4: The year on the Roman et reference is given as 2015 in the text, but is listed as 2016 in the reference list. Please make consistent to the proper year.

**Updated to 2016 in the text.**

Pg. 19, line 8: The year on the Arguez et al reference is given as 2010 in the text, but is listed as 2012 in the reference list. Please make consistent to the proper year.

**Updated to 2012**